# Fungal kinases and transcription factors regulating brain infection in *Cryptococcus neoformans*

Kyung-Tae Lee [1,6], Joohyeon Hong [1,6], Dong-Gi Lee [1,6], Minjae Lee [1], Suyeon Cha[1], Yu-Gyeong Lim [1], Kwang-Woo Jung[2], Areum Hwangbo[1], Yelin Lee [1], Shang-Jie Yu[3,5], Ying-Lien Chen [3], Jong-Seung Lee [4], Eunji Cheong [1✉] & Yong-Sun Bahn [1✉]

*Cryptococcus neoformans* causes fatal fungal meningoencephalitis. Here, we study the roles played by fungal kinases and transcription factors (TFs) in blood-brain barrier (BBB) crossing and brain infection in mice. We use a brain infectivity assay to screen signature-tagged mutagenesis (STM)-based libraries of mutants defective in kinases and TFs, generated in the *C. neoformans* H99 strain. We also monitor in vivo transcription profiles of kinases and TFs during host infection using NanoString technology. These analyses identify signalling components involved in BBB adhesion and crossing, or survival in the brain parenchyma. The TFs Pdr802, Hob1, and Sre1 are required for infection under all the conditions tested here. Hob1 controls the expression of several factors involved in brain infection, including inositol transporters, a metalloprotease, *PDR802*, and *SRE1*. However, Hob1 is dispensable for most cellular functions in *Cryptococcus deuterogattii* R265, a strain that does not target the brain during infection. Our results indicate that Hob1 is a master regulator of brain infectivity in *C. neoformans*.

--------

[1] Department of Biotechnology, College of Life Science and Biotechnology, Yonsei University, Seoul 03722, Korea. [2] Research Division for Biotechnology, Korea Atomic Energy Research Institute, Jeongeup 56212, Korea. [3] Department of Plant Pathology and Microbiology, National Taiwan University, Taipei 10617, Taiwan. [4] AmtixBio Co., Ltd., Seoul 05836, Korea. [5] Present address: National Institute of Infectious Diseases and Vaccinology, National Health Research Institute, 35053 Miaoli, Taiwan. [6] These authors contributed equally: Kyung-Tae Lee, Joohyeon Hong, Dong-Gi Lee. ✉email: eunjicheong@yonsei.ac.kr; ysbahn@yonsei.ac.kr

In the past decades, there has been a growing concern about fungal pathogens that impose a serious threat to both animal and plant ecosystems[1]. A recent epidemiological study has demonstrated that more than 1.5 million people succumb to death due to invasive mycoses[2]. *Cryptococcus neoformans* is one of the major invasive mycosis-causing opportunistic fungal pathogens, which is responsible for more than 220,000 infections and 180,000 deaths globally every year[3]. Nevertheless, treatment options are highly limited and mortality rate is unacceptably high. Therefore, it is imperative to fully understand the signalling and metabolic networks governing the pathogenicity of *C. neoformans* during host infection for the development of antifungal targets and drugs.

Pathogenic *Cryptococcus* species complex is widely distributed in a variety of environmental niches[4]. In the natural environment, it generates infectious propagules through heterosexual or unisexual reproduction, which can be inhaled through the respiratory tract of a susceptible mammalian host. At this initial stage of infection, *C. neoformans* battle against alveolar macrophages residing in the lungs by employing antiphagocytic factors, such as capsule and melanin pigment[5–8]. Even after phagocytosis, the pathogen can survive and proliferate within the phagolysosome and escape from the macrophage, often even without killing the host cell[9–11]. Subsequently, *C. neoformans* haematogenously disseminates and eventually travels to the central nervous system (CNS), causing meningoencephalitis. Among the infection stages, crossing the blood-brain barrier (BBB) and proliferation in the brain parenchyma are critical factors for *C. neoformans* to impose lethal lesions in mammalian brain tissues. *C. neoformans* is able to traverse the BBB, which protects the brain from pathogens because of the establishment of tight junctions between brain microvascular endothelial cells (BMECs) and the contribution of brain astrocytes[12], through transcellular, paracellular, and/or Trojan-horse mechanisms[13,14]. Nevertheless, the complex signalling pathways regulating these known and hitherto uncharacterised factors governing the BBB crossing and brain infection of *C. neoformans* are not completely understood.

To systematically dissect infection-stage dependent signalling networks, here we performed signature-tagged mutagenesis (STM)-based brain infectivity assay with transcription factor (TF) and kinase mutant libraries using intravenous infection route, which can bypass lung infection, and compared these data with that of previously acquired lung-STM data using the same set of mutant libraries but intranasally infected[15,16]. In addition, we performed in vivo transcription profiles of 180 TFs and 183 kinases, along with 58 known virulence-related genes, by using NanoString technology at diverse infected tissues during *Cryptococcus* infection. For brain-infection-related TFs and kinases, we further analysed their roles in BBB adhesion/crossing and survival within the brain parenchyma. Combining these systematic analyses, we not only provide a comprehensive insight into how different signalling networks are modulated in vivo during the different stages of *C. neoformans* infection, but also unravelled a large number of TFs and kinases involved in the brain infection of the pathogen. Most importantly, we demonstrate that the homeobox TF Hob1 is the master regulator for a number of brain-infection-related genes in the *C. neoformans* H99 strain, but not in the *Cryptococcus deuterogattii* R265 strain that does not target the brain during infection.

## Results

**Brain infectivity assay with *C. neoformans* kinase and TF mutants.** We previously performed STM-based murine lung infectivity assays with the kinase and TF mutant libraries (264 strains representing 129 kinases and 322 strains representing 155 TFs) using the lungs recovered after 14 days post infection (dpi) from intranasally infected mice and identified 58 and 40 lung infectivity-related kinases[15] and TFs[16], respectively. These may represent the signalling components required for the initial stage of cryptococcal infection. To identify the signalling components required for the brain infection, which represents the late stage of cryptococcal infection, the STM scores of TF and kinase mutants recovered from the brain tissues of the same infected mice were measured (Supplementary Fig. 1). However, we were often unable to recover a sufficient amount of *C. neoformans* cells from the brain tissues of some mice for STM analysis. In addition, as the intranasally infected *C. neoformans* mutants should pass through the lungs first, those reaching the brain were no longer equally distributed unlike pooled input mutants, rendering direct comparison of the lung and brain-STM scores difficult. Therefore, we performed another round of STM-based murine infectivity assay for the TF and kinase mutant libraries using the brain recovered from intravenously infected mice after 7 dpi (see the strategy of Fig. 1a) and compared the brain-STM score with the previous lung-STM score for each mutant (Fig. 1b, c; Supplementary Fig. 2).

Regarding kinases, the 34 kinases that were found to be required for both the lung and brain infections were defined as core-virulence kinases. Some of them were signalling components of previously characterised signalling pathways in *C. neoformans*: Pka1 in the cAMP signalling pathway[17,18], Ssk2 and Hog1 in the high osmolarity glycerol response (HOG) pathway[19,20], Bck1 and Mpk1 in the cell wall integrity MAPK pathway[21,22], Ire1 in the unfolded protein response (UPR) pathway[23], Vps15 in the vacuole-trafficking pathway[15], Snf1 and Gal83 in the carbon utilisation pathway[15], Bud32 in the KEOPS/EKC complex, Ypk1, Gsk3, and Ipk1 in the the TOR (Target of Rapamycin) pathway[15]. Except for these known proteins, the core virulence-related kinases also include Irk2 and Irk5, whose regulatory mechanisms are not evident. Irk2 and Irk5 belong to the families of APH phosphotransferases, diacylglycerol kinase-like kinase and AGC/YANK protein kinase, respectively. Deletion of *IRK5* significantly reduces melanin production, but enhances capsule production[15], indicating that defective melanin formation may be attributable to the role of Irk5 in virulence. However, deletion of *IRK2* did not result in evident in vitro phenotypic alteration. In contrast to the 34 core-virulence kinases, 23 and 12 kinases were required only for the lung or brain infection, respectively (Fig. 1b; STM cutoff >2.0 or < −2.0; *P* < 0.05).

In the case of TFs, a total of 9 TFs were found to be required for both lung and brain infections and here defined as core-virulence TFs. These include Sre1 and Hob1 in the sterol biosynthesis pathway[16], Hxl1 in the UPR pathway[23], and Gat201 and Nrg1 in the capsule biosynthetic pathway[24,25]. Besides these known TFs, Pdr802, Fzc1, Fzc9 and Fzc31, which all contain the fungal specific $Zn_2Cys_6$ DNA binding domain, were also identified as core-virulence TFs. Notably, in vivo functions of these four TFs attributed to the pathogenicity of *C. neoformans* were not evident based on their in vitro mutant phenotypic traits[16]. Deletion of *FZC1* reduced the growth at 39 °C (but not at 37 °C) and mating capacity but increased capsule and melanin production. Deletion of *FZC9* reduced mating and resistance to hydrogen peroxide. Deletion of *FZC31* reduced the growth under high temperature and oxidative stresses and mating capacity but increased melanin production. In contrast to the 9 core-virulence kinases, 23 and 10 TFs were required only for the lung or brain infection, respectively (Fig. 1c; STM cutoff >2.0 or < −2.0; *P* < 0.05).

Overall, deletion of kinase genes resulted in more dramatic changes in both the lung and brain-STM scores than deletion of TFs did, probably because kinases generally function upstream of

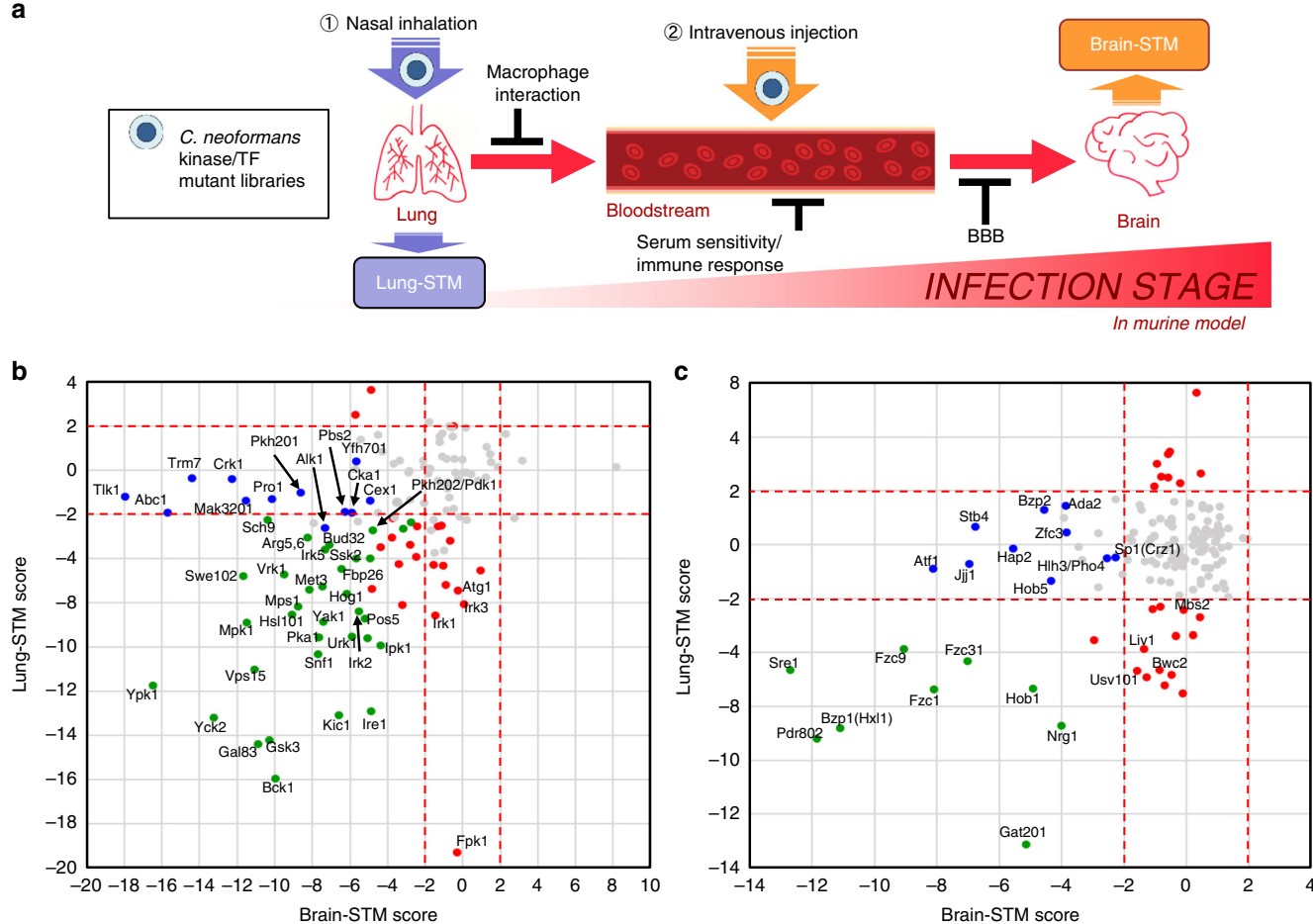

**Fig. 1 Dual signature-tagged mutagenesis (STM) analyses of kinase and TF deletion libraries of *C. neoformans*.** (**a**) A schematic diagram for comparison of lung (via intranasal infection) and brain (via intravenous infection) STM analysis using *C. neoformans* TF and kinase mutant libraries[15, 16]. (**b-c**) Lung-STM scores for each mutant were obtained from the previous reports using a intranasally infected murine model (A/J mice, *n* = 3). Brain-STM scores were obtained in this study using the A/J mouse model intravenously infected with the same set of TF and kinase mutant libraries (the whole data were shown in Supplementary Fig. 2). Dual STM scores of (**b**) 129 kinases and (**c**) 155 TFs were represented by brain-STM score (X-axis) and lung-STM score (Y-axis). Each spot was an average STM score of two-independent TF or kinase mutant strains, each of which was obtained from three mice (*n* = 3). We used the colour code for kinase/TF mutants that exceeded STM cutoff [>2.0 (high) or < −2.0 (low)] and had statistically significant *P* values (<0.05) by one-way ANOVA analysis with Bonferroni's multiple comparison test. Green dots indicate the core-virulence kinase/TF mutants that show low/high STM scores in both lung- and brain infections. Blue dots indicate the kinase/TF mutants that show low/high STM scores only in brain-infection. Red dots indicate the kinase/TF mutants that show low/high STM only in lung infection.

TFs in signalling pathways. These results clearly indicate that redundant and distinct signalling components are involved in different cryptococcal infection stages.

**In vivo transcription profiles of kinases and TFs during infection.** To further gain insight into the in vivo regulatory mechanisms of pathogenicity-related kinases and TFs at different *C. neoformans* infection stages, we performed NanoString nCounter-based in vivo transcription analysis of 180 TFs and 183 kinases, along with 58 known virulence-related genes (See Supplementary Data 1). This platform has been effectively used to quantitatively measure in vivo gene expression of target genes of a pathogen during host infection[26]. For this analysis, mice were intranasally infected with the *C. neoformans* H99 strain and infected lung, brain, kidney, and spleen tissues were recovered after 3 (early stage), 7 (middle stage), 14, and 21 dpi (late stages) (Fig. 2a). Total murine and cryptococcal RNAs were isolated from each infected tissue and used for NanoString assays to quantitate pathogen-specific mRNA transcripts using designed probes by normalisation with average expression levels of eight

housekeeping genes (Supplementary Data 1). In vivo expression levels of each target gene at different tissues and days of infection were compared to their in vitro expression levels under basal growth conditions [yeast extract-peptone-dextrose (YPD) medium at 30 °C].

We first checked how 58 known virulence-related genes are differentially regulated during infection. Among these, genes involved in metal ion sensing and uptake, such as *CIG1* and *CFO1*, were highly upregulated at different tissues at all infection stages (Fig. 2b), further supporting that Cig1 and Cfo1 are essential for virulence of *C. neoformans*[27,28]. The copper regulon genes such as *CnMT1/2* (metallothioneins) and *CTR4* (copper transporter) were also highly upregulated at all infected tissues from the early to the late infection stages. In addition, genes involved in the production of two major virulence factors, melanin (*LAC1*) and capsule (*CAP10, CAP59, CAP60, CAP64*), were differentially regulated during infection. Notably, in vivo expression levels of *LAC1* increased during 3~14 dpi and decreased at 21 dpi, implying that *LAC1* induction is associated with the survival and adaptation of *C. neoformans* in the host.

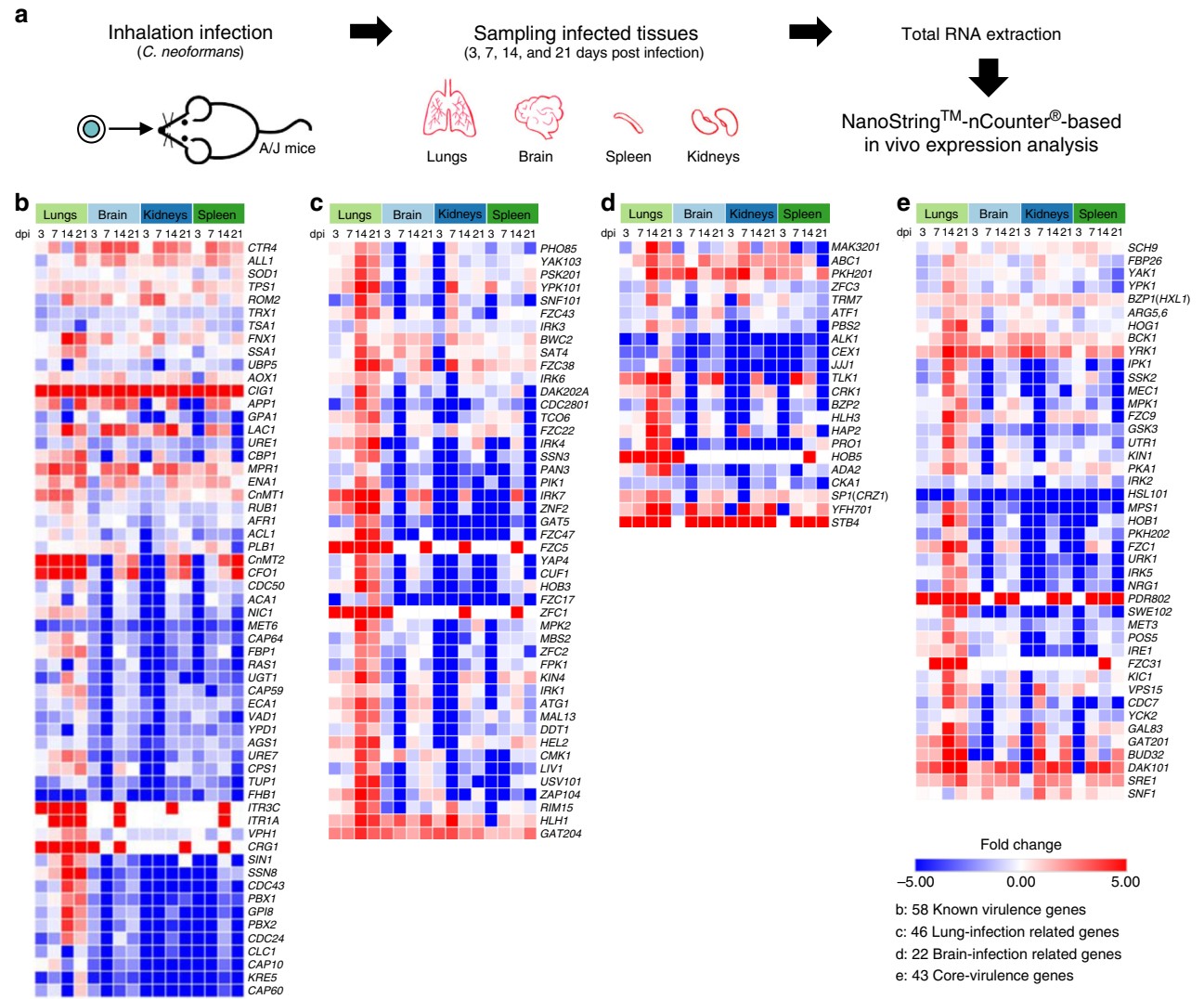

**Fig. 2 In vivo expression profiling of *C. neoformans* transcription factor and kinase genes in comparison with known cryptococcal virulence genes.** (**a**) Graphical abstract of NanoString™-nCounter® based in vivo gene expression analysis in a murine model of systemic cryptococcosis through intranasal instillation. Total RNA was extracted from the infected lung, spleen, kidney, and brain tissues recovered after 3, 7, 14, and 21 dpi. Three mice per cohort were used (total 12 mice). Gene-specific nCounter codesets were designed for 58 published virulence-related genes, 183 kinases, 180 TFs, and 8 housekeeping genes for expression normalisation (Supplementary Data 1 and Supplementary Fig. 3). (**b-e**) Fold-change heatmap of in vivo gene expression of (**b**) 58 published (known) virulence-related genes, (**c**) 42 lung-infection-related genes, (**d**) 22 brain-infection-related genes, and (**e**) 43 core-virulence genes.

Unexpectedly, in vivo expression of *CAP10*, *CAP59*, *CAP60*, and *CAP64* was generally downregulated in all infected tissues except the lungs where they were upregulated by about 2 fold between 7 to 21 dpi (Fig. 2b).

The in vivo transcription profiling analysis revealed the following expression patterns of 183 kinase and 180 TF genes during the whole infection process. Reflecting that the lungs are the initial infection sites for *C. neoformans*, expression of a large number of kinases and TF genes were induced in the lungs, particularly after 14 dpi (Supplementary Fig. 3). Notably, we found that no genes were specifically expressed only in the spleens, kidneys, or brains. Collectively, these results indicated that transcriptional modulation of pathogenicity-related kinases and TFs appeared to be initiated at the lung infection stage of cryptococcal infection.

**Kinases and TFs required for adhering and crossing the BBB.** We further focused on the 12 kinase and 10 TF mutants that

showed significantly reduced brain-STM score (cutoff < −2.0; *P* < 0.05), but not in the lung (Fig. 1b, c and Supplementary Data 2). The change in the brain-STM score could occur during haematogenous dissemination. Therefore, we monitored the serum sensitivity of these TF and kinase mutants in the presence of 50% foetal bovine serum (FBS) in YPD medium. Surprisingly, only the *pho4Δ* mutant (also known as *hlh3Δ*) showed marked serum-specific growth defect (Supplementary Fig. 4), which is in agreement with the previously reported data[29]. Therefore, the low brain-STM score of the *pho4Δ* mutant, at least partly, resulted from its serum sensitivity.

For the remaining low brain-STM TF and kinase mutants, we hypothesised that some of these mutants could have altered the ability to traverse the BBB. Previously, the in vitro BBB system, which consists of human brain microvascular endothelial cells (HBMECs) growing on a transwell membrane separating the top compartment (blood site) and bottom compartment (brain side), had been constructed by Vu et al.[30] and successfully employed to

discover BBB-traversal factors, including Mpr1[31]. In this study, we similarly established the transwell-based in vitro BBB system and tested the capability of the selected brain-infection-related TF and kinase mutants in traversing the BBB. To verify our in vitro BBB system, we first compared the BBB-traversal ability of *C. neoformans* (H99 strain) with non-pathogenic *S. cerevisiae* (S288C) and the *mpr1Δ* mutant (Supplementary Fig. 5) that was independently constructed in this study. After 24 h incubation, about 10% of the wild-type (WT) *C. neoformans* were able to traverse the BBB (Fig. 3a). In contrast, *S. cerevisiae* did not traverse the BBB at all and the *C. neoformans mpr1Δ* mutant was highly defective in BBB traversal (less than 2%) (Fig. 3a). Similar to the previous finding[32], we did not find any significant changes in trans-endothelial electrical resistance (TEER) (Fig. 3a), further verifying that the tight junction joining HBMECs was not affected during the BBB traversal of *C. neoformans* cells.

We next assessed the 12 kinase and 10 TF mutants, which showed low STM score in the brain, but not in the lung, for their capability to traverse the BBB. Among these, we excluded the *cka1Δ*, *pro1Δ*, and *bzp2Δ* mutants in this analysis, as they exhibit significantly retarded growth at 37 °C[15,16]. We found that a total of 5 kinases (Cex1, Alk1, Pbs2, Yfh701, and Pkh201) and 4 TFs (Ada2, Hap2, Pho4, and Jjj1), whose deletion all resulted in low brain-STM scores, were required for the BBB traversal (Fig. 3b). As the first step for the BBB traversal of *C. neoformans* is its adhesion to the surface of the HBMECs, we further examined the capability of the BBB-crossing defective 5 kinase and 4 TF mutants to adhere to the monolayer of HBMECs. Among these, *cex1Δ*, *pbs2Δ*, *alk1Δ*, *pkh201Δ*, and *hap2Δ* mutants showed reduced adhesion to HBMECs (Fig. 3c), indicating that host cell adhesion is an important pre-requisite for efficient BBB crossing of *C. neoformans* (summarised in Supplementary Data 2).

We also examined the BBB crossing ability of core-virulence 34 kinase and 9 TF mutants except those defective in growth at 37 °C (*yck2Δ*, *arg5,6Δ*, *fbp26Δ*, *mps1Δ*, *mec1Δ*, *swe102Δ*, *cdc7Δ*, *gsk3Δ*, *bud32Δ*, *kic1Δ*, *mpk1Δ*, *bck1Δ*, *pdk1/pkh202Δ*, *utr1Δ*, *pos5Δ*, *ipk1Δ*, *ire1Δ*, *vps15Δ*, *ypk1Δ*, *nrg1Δ*, and *hxl1Δ*). Among the remaining 15 kinase and 7 TF candidates, *hog1Δ*, *ssk2Δ*, *pka1Δ*, and *gat201Δ* mutants were as efficient in BBB crossing as WT strain, suggesting that Hog1 MAPK, cAMP/PKA, and Gat201-dependent pathways are not involved in BBB crossing (Fig. 3d). Interestingly, the finding that the Pbs2 MAPK kinase (MAPKK), but not its downstream Hog1 MAPK, was involved in BBB crossing indicated that Pbs2 may have downstream targets other than Hog1 for regulating BBB crossing. In contrast, Sre1 and Hob1 in the sterol biosynthesis pathway, Sch9 in the TOR pathway, Met3 in the sulphur assimilation pathway, Snf1 and Gal83 in the carbon utilisation pathway, and Vrk1 in ribosome biogenesis pathway appear to promote BBB crossing of *C. neoformans* (Fig. 3d). In addition to these kinases and TFs with known pathways, Hsl101, Irk2, Irk5, Urk1, Fzc1, Fzc9 and Pdr802 in undefined pathways were also found to be critical for BBB crossing. Furthermore, the *met3Δ*, *snf1Δ*, *vrk1Δ*, *gal83Δ*, *hsl101Δ*, *irk2Δ*, *sre1Δ*, *pdr802Δ*, *fzc9Δ*, *hob1Δ*, and *fzc1Δ* mutants, but not *irk5Δ*, *sch9Δ*, and *urk1Δ* mutants, showed reduced adhesion to the monolayer of HBMECs, further supporting that host cell adhesion is a critical pre-requisite for efficient BBB crossing of *C. neoformans* (Fig. 3e). Collectively, these data suggest that *C. neoformans* employs complex signalling networks involved in a variety of biological processes to cross the BBB (summarised in Supplementary Data 3).

Notably, however, the NanoString-based in vivo transcription profiling analysis revealed that these brain infectivity-related kinases and TFs were not specifically modulated only in the brain. In fact, most of these genes were upregulated in the lungs at the later stage of infection (14 and 21 dpi) (Fig. 3f). Particularly,

in vivo expression of *PDR802*, *SRE1*, *VRK1*, *PKH201*, and *YFH701* was strongly induced in all infected tissues tested during almost all infection stages (Fig. 3f). In contrast, regardless of the critical roles of Cex1 and Met3 in BBB crossing and adhesion to HBMECs, their in vivo expression levels were not strongly induced at all infection stages and in all tissues (Fig. 3f).

**Kinases and TFs required for survival in the brain parenchyma.** A question remains for the TF and kinase mutants, which were normal in haematogenous dissemination and crossing the BBB, but still showed significant changes in brain-STM score. We hypothesised that some of these mutants may lack an ability to survive in the brain parenchyma even after crossing the BBB. To address this question, we directly monitored the capability of the low brain-STM TF and kinase mutants to proliferate in the brain parenchyma. For this purpose, we established an intracerebroventricular (ICV) method of infection of the mouse brain with *C. neoformans* by bypassing the BBB (Fig. 4a), which allowed us to infect the brain parenchyma consistently and with an equal number of cryptococcal cells. Once we infected a group of mice with the low brain-STM TF and kinase mutants by ICV injection, we recovered mutants from the infected brain after 6 dpi and assessed the STM score by qPCR (here abbreviated as ICV-STM score). We found that 6 strains with deleted kinases (*TLK1*, *TRM7*, *CRK1*, *MAK3201*, *PKH201*, or *ALK1*) and 4 with deleted TFs (*ADA2*, *BZP2*, *ZFC3*, or *HAP2*) displayed significantly reduced ICV-STM score (Fig. 4b,c). Among these, 2 kinases (Pkh201 and Alk1) and 2 TFs (Ada2 and Hap2) were required for both BBB crossing and survival in the brain parenchyma. Notably, the kinases and TFs that were required for BBB crossing (Cex1, Pbs2, Yfh701, Pho4, and Jjj1) were dispensable for proliferation in the brain parenchyma (Fig. 4b, c). In contrast, 4 kinases (Tlk1, Trm7, Crk1, and Mak3201) and a single TF (Zfc3) were uniquely involved in proliferation in the brain parenchyma, but not in BBB crossing (Fig. 4b, c).

We also monitored ICV-STM score for 30 and 9 core-virulence kinase and TF mutants. As expected, most kinase and TF mutants that were defective in growth at 37 °C (*yck2Δ*, *arg5,6Δ*, *mpk1Δ*, *mps1Δ*, *kic1Δ*, *bud32Δ*, *bck1Δ*, *utr1Δ*, *fbp26Δ*, *pos5Δ*, *mec1Δ*, *ipk1Δ*, *swe102Δ*, and *nrg1Δ* mutants) also showed reduced ICV-STM score (Fig. 4d, e). Among the core-virulence kinases and TFs that were required for BBB crossing, Irk2, Vrk1, Pdr802, Sre1, and Hob1 were also required for proliferation in the brain parenchyma (Fig. 4d, e). Interestingly, the *hog1Δ* mutant showed low ICV-STM score (Fig. 4d), indicating that Hog1 is not required for BBB crossing, but required for proliferation in the brain parenchyma. In contrast, Sch9, Irk5, Fzc1, and Fzc9 were dispensable for survival in the brain parenchyma (Fig. 4d, e). In addition, we found that Snf1, Gal83, Kin1, Urk1, Hsl101, Yak1, Fbp26, Pos5, and Swe102 were involved in the proliferation in the brain parenchyma. Collectively, our data clearly indicated that *C. neoformans* employs redundant and distinct sets of signalling pathways to cross the BBB and proliferate in the brain parenchyma.

**Hob1 is a master regulator of brain infectivity in *C. neoformans*.** We next aimed to address how regulatory networks of known brain-infection factors, such as Ure1, Cps1, Itr1a, Itr3c, Mpr1, Plb1, Fnx1, and Rub1, are connected to brain-infection-related kinases and TFs identified in this study. To this end, first we examined whether the known brain-infection factors are transcriptionally regulated by in vitro host-mimic conditions (HMC): a tissue culture medium (RPMI) supplemented with 10% FBS at 37 °C under 5% $CO_2$. We found that expression of *ITR1a*, *ITR3c*, and *MPR1* was strongly induced (~3–4 folds) by HMC

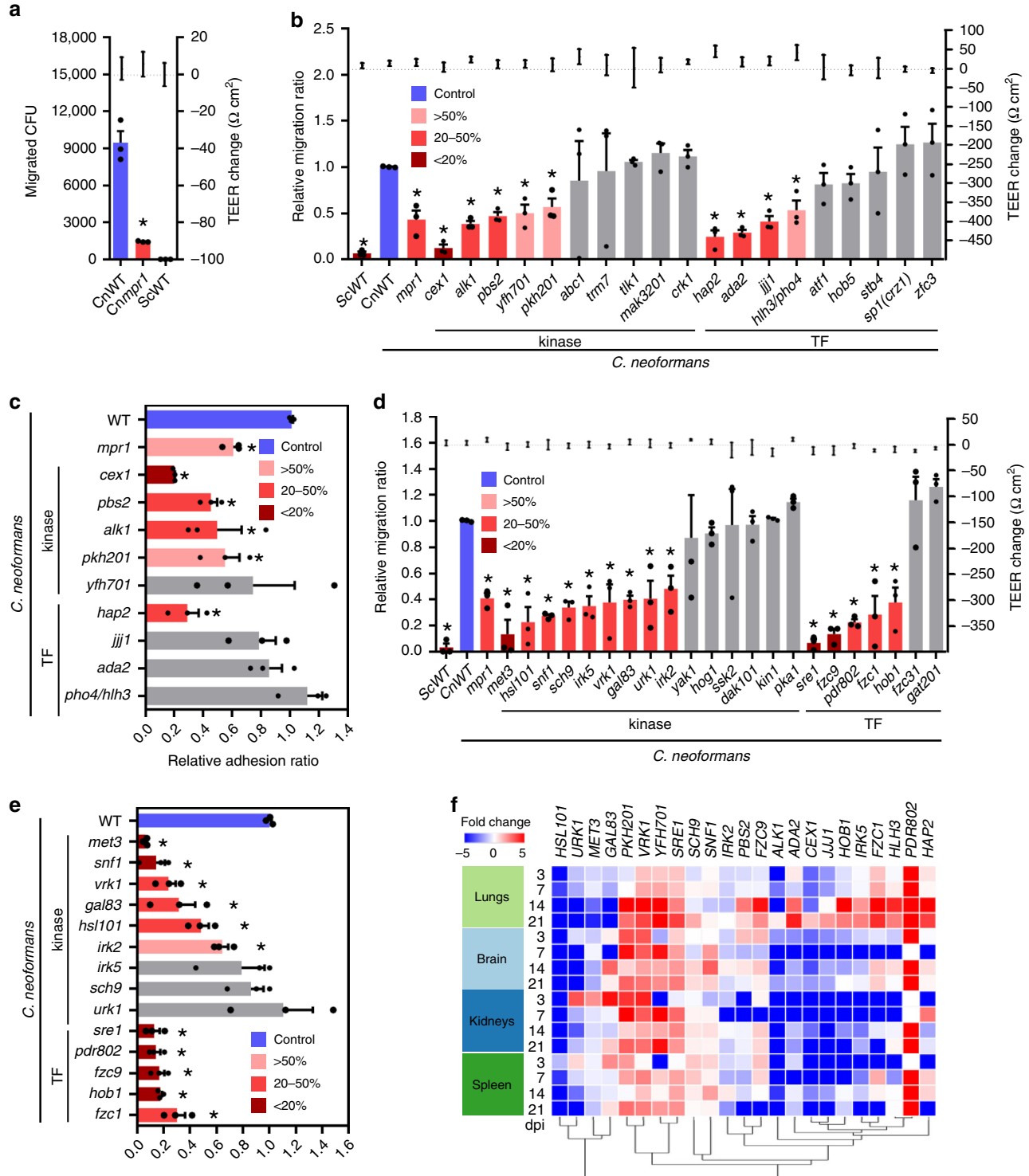

**Fig. 3 In vitro BBB transmigration and adhesion assays for *C. neoformans* transcription factor and kinase mutants having low brain STM scores.** BBB transmigration assays with hCMEC/D3-coated Transwell systems (**a**, **b**, **d**) and hCMEC/D3 cell adhesion assays (**c**, **e**). **a**, **b**, **d** Plates with inoculum of $10^5$ yeast cells were incubated at 37 °C in $CO_2$ incubator for 24 h, and the number of yeast cells passing through the hCMEC/D3-coated Transwell was measured by CFU. *S. cerevisiae* WT (*Sc*WT, S288C) and *C. neoformans* *mpr1*Δ mutant (*Cnmpr1*) were used as negative controls. **a** Left and right *Y* axes indicate the migrated CFU and the trans-endothelial electrical resistance (TEER) value, respectively. **b**, **c** The results of BBB transmigration (**b**) and adhesion assays (**c**) of brain-infection-related kinases and TFs discovered by brain-STM analysis. Left and right *Y* axes indicate the relative migration ratio and the TEER value, respectively. **d**, **e** The results of BBB transmigration (**d**) and cell adhesion assay (**e**) of core-virulence kinases and TFs discovered by dual STM analyses. Each mutant was analysed by more than three biologically independent experiments with three technical replicates. List of strains used in this Figure is described in Supplementary Data 4. **a**–**e** Data are presented as mean values ± standard error of the mean (SEM). Statistical significance of difference between *Cn*WT (H99 strain) and each mutant (*, *P* < 0.05) was calculated by two-tailed Student's *t* test (unpaired, *n* = 3). **f** NanoString heatmap of BBB-crossing-related kinases and TFs was clustered by using Morpheus.

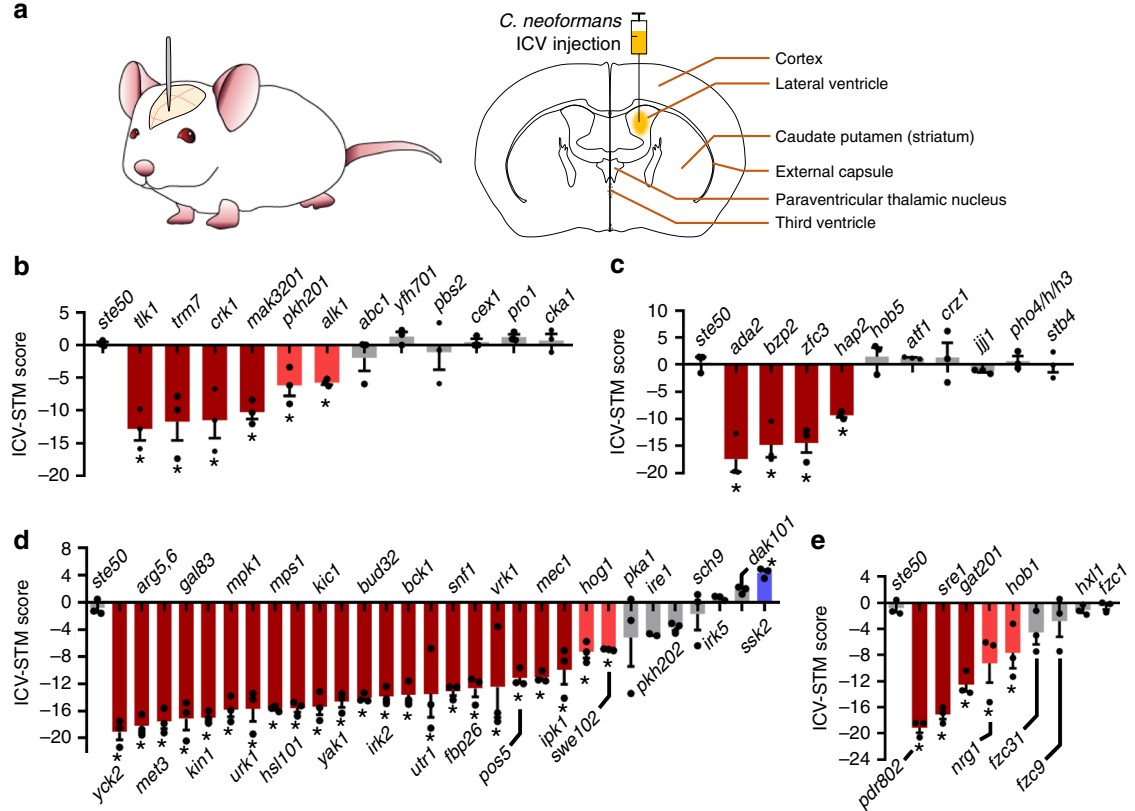

**Fig. 4 Identification of genes required for survival of *C. neoformans* in the brain parenchyma through intracerebroventricular (ICV)-STM analysis.** (**a**) Graphical abstract of ICV injection for directly infecting the brain parenchyma with *C. neoformans*. (**b**–**e**) ICV-STM score of brain-infection-related (**b**) kinases, (**c**) TFs, core-virulence (**d**) kinases, and (**e**) TFs. Three mice (A/J) were infected with each pooled mutant group (Supplementary Data 4), sacrificed after 7 dpi, and infected brain tissues were harvested for recovering cryptococcal cells. The ICV-STM score was calculated by quantitative PCR of genomic DNA isolated from input and output cryptococcal mutants as described in Methods. The *ste50Δ* mutant was used as a virulent control strain. Red and blue marks (with *) indicate mutants whose STM scores are statistically different (*P* < 0.05, *n* = 3) from the *ste50Δ* STM score by one-way ANOVA analysis with Bonferroni's multiple comparison test. Data are presented as mean values ± SEM.

(Fig. 5a), further indicating that inositol transporters and metalloprotease are critical for BBB crossing. We also monitored whether expression of any BBB-crossing-related TFs that were discovered in this study were also induced by HMC. Except for *PHO4*, none of brain-infection-related TFs described in Fig. 3b (*ADA2*, *HAP2*, and *JJJ1*) were induced by HMC (Fig. 5b). We then paid attention to some of the core-virulence TFs that were required for both lung and brain infections: Fzc1, Fzc9, Fzc31, Pdr802, Hob1, Sre1, Hxl1, Nrg1, and Gat201 (Fig. 1c). Expression of *FZC31*, *PDR802*, *HOB1*, *SRE1*, *HXL1*, and *GAT201* was induced by HMC, whereas expression of *FZC9* and *NRG1* was rather repressed (Fig. 5c).

Among these, we further focused on Pdr802, Hob1, and Sre1, because their expression was induced by HMC and they were required for both BBB crossing and proliferation in the brain parenchyma. Moreover, the in vivo expression of *PDR802* and *SRE1* was strongly induced at all infected tissues throughout the whole infection stages, although *HOB1* expression was induced at the lungs only after 14 dpi (Fig. 5d). Therefore, we hypothesised that these TFs could be potential candidates for regulating HMC-inducible genes. To test this hypothesis, we assessed the expression levels of HMC-inducible genes in *sre1Δ*, *hob1Δ*, and *pdr802Δ* mutants in comparison with those of the WT strain. Surprisingly, HMC-mediated induction of *ITR1a*, *ITR3c*, and *MPR1* was markedly reduced in the *hob1Δ* mutant, but not in the *pdr802Δ* and *sre1Δ* mutants (Fig. 5e). Notably, even HMC-induced *PDR802* and *SRE1*, along with *FZC31*, were significantly

reduced in the *hob1Δ* mutant (Fig. 5e). In contrast, *GAT201* was induced normally by HMC in the *hob1Δ* mutant, suggesting that *GAT201* expression may be controlled by other TFs. The HMC-mediated induction of *FZC31* was also regulated by Sre1 as well as Hob1 (Fig. 5e). Collectively, these data indicate that Hob1 is a key regulator of induction of many brain-infection-related genes.

**Hob1 is dispensable for the pathogenicity of *C. deuterogattii*.** The critical role of Hob1 in the regulation of brain-infection-related genes in *C. neoformans* prompted us to question its role in its pathogenic *Cryptococcus* sibling species *C. deuterogattii*, which is known to mainly target the lungs, but not the brain during infection[33,34]. Among various *C. deuterogattii* strains, here we focused on the R265 strain, which belongs to the VGIIa clade and is most widely used for pathobiological study of *C. gattii* species complex[35,36]. The R265 strain contains a single conserved Hob1 orthologue in its genome (score: 1015, e-value: 0.0). Therefore, we constructed the *hob1Δ* mutants in the R265 strain background (Supplementary Fig. 6) and compared their phenotypes with those of H99 *hob1Δ* mutants. The H99 *hob1Δ* mutants exhibited thermosensitivity, defects in melanin production, increased sensitivity to osmotic shock, oxidative stress, genotoxic stress, cell membrane/wall stress, and defective virulence[16] (Fig. 6a and Supplementary Fig. 8). Notably, the *hob1Δ* mutants displayed high resistance to fluconazole, but increased susceptibility to amphotericin B, suggesting the role of Hob1 in ergosterol biosynthesis. Re-integration of the WT copy of *HOB1* into its native

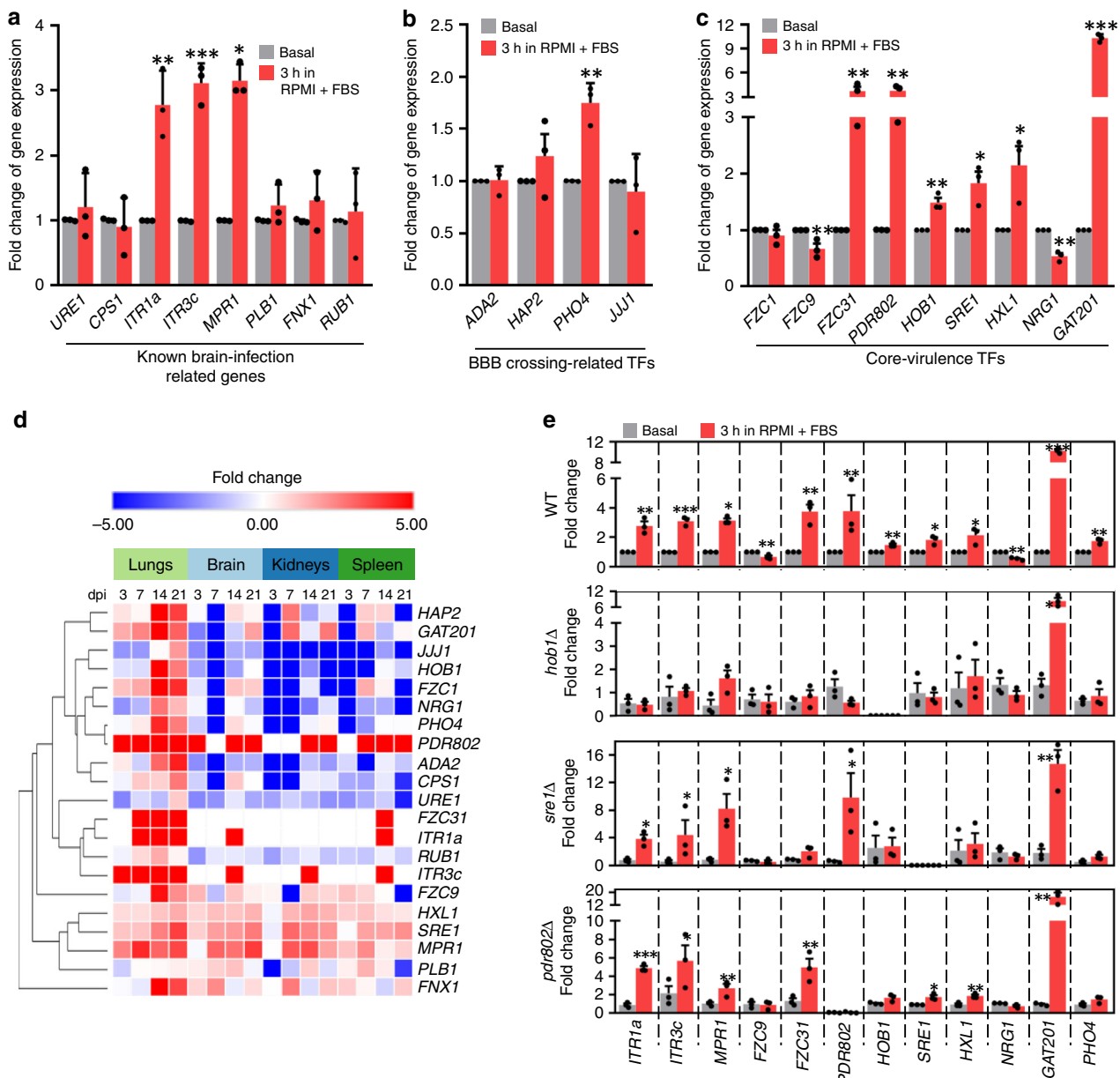

**Fig. 5 The homeobox transcription factor Hob1 as a master regulator of brain-infection-related genes in *C. neoformans*. a–c, e** Host-mimic condition (HMC) mediated induction of known brain-infection-related genes (**a**), BBB crossing-related TFs (**b**), and core-virulence TFs (**c**). Expression of each gene was determined by quantitative RT-PCR with cDNA synthesized from total RNA isolated from *C. neoformans* WT (H99) and mutant (*hob1*Δ, *sre1*Δ and *pdr802*Δ) strains that were shifted from basal condition (YPD at 30 °C; control samples; grey bars) to HMC (RPMI with 10% foetal bovine serum at 37 °C under 5% $CO_2$) and incubated for 3 h (red bars). Fold change of gene expression was calculated relative to the basal expression levels of each gene. Data from three independent experiments performed in duplicates are presented as mean values ± SEM. Statistical significance of difference in gene expression levels between basal and HMC condition (unpaired, $n = 3$) was determined by two-tailed Student's $t$ test (*, $P < 0.05$; **, $P < 0.001$; ***, $P < 0.0001$). **d** NanoString heatmap of genes used in (**a**-**c**) was clustered by using Morpheus. Bold letters indicated the genes whose expression level was induced or reduced under HMC.

locus restored WT phenotypes in the *hob1*Δ mutants, verifying the function of Hob1 (Supplementary Fig. 10). Deletion of *HOB1* in *C. deneoformans* XL280 strain background resulted in similar phenotypic changes, albeit to a lesser extent than those in H99 strain background (Fig. 6a). Strikingly, however, *HOB1* deletion did not cause any significant phenotypic changes in R265 strain background (Fig. 6a and Supplementary Fig. 8). In contrast, deletion of *SRE1*, which is a major sterol-regulating TF in the SREB pathway, resulted in similar phenotypic changes, as shown by the highly increased susceptibility to fluconazole, of

both H99 and R265 strains (Fig. 6a and Supplementary Fig. 8), suggesting that Sre1 plays conserved roles in both strains.

R265 Hob1 with a shorter N-terminal extension domain (558 amino acids) exhibited 91.9% identity to H99 Hob1 (591 amino acids) (Supplementary Fig. 9a). Their homeobox domains, which are located at the N-terminus, have two amino acid changes (Supplementary Fig. 9a). To address whether such differences in Hob1 sequences are responsible for their distinct functions, we constructed *hob1*Δ::*HOB1*[R265] allele-swapping strains, in which the R265 *HOB1* allele was integrated into the native *HOB1* locus

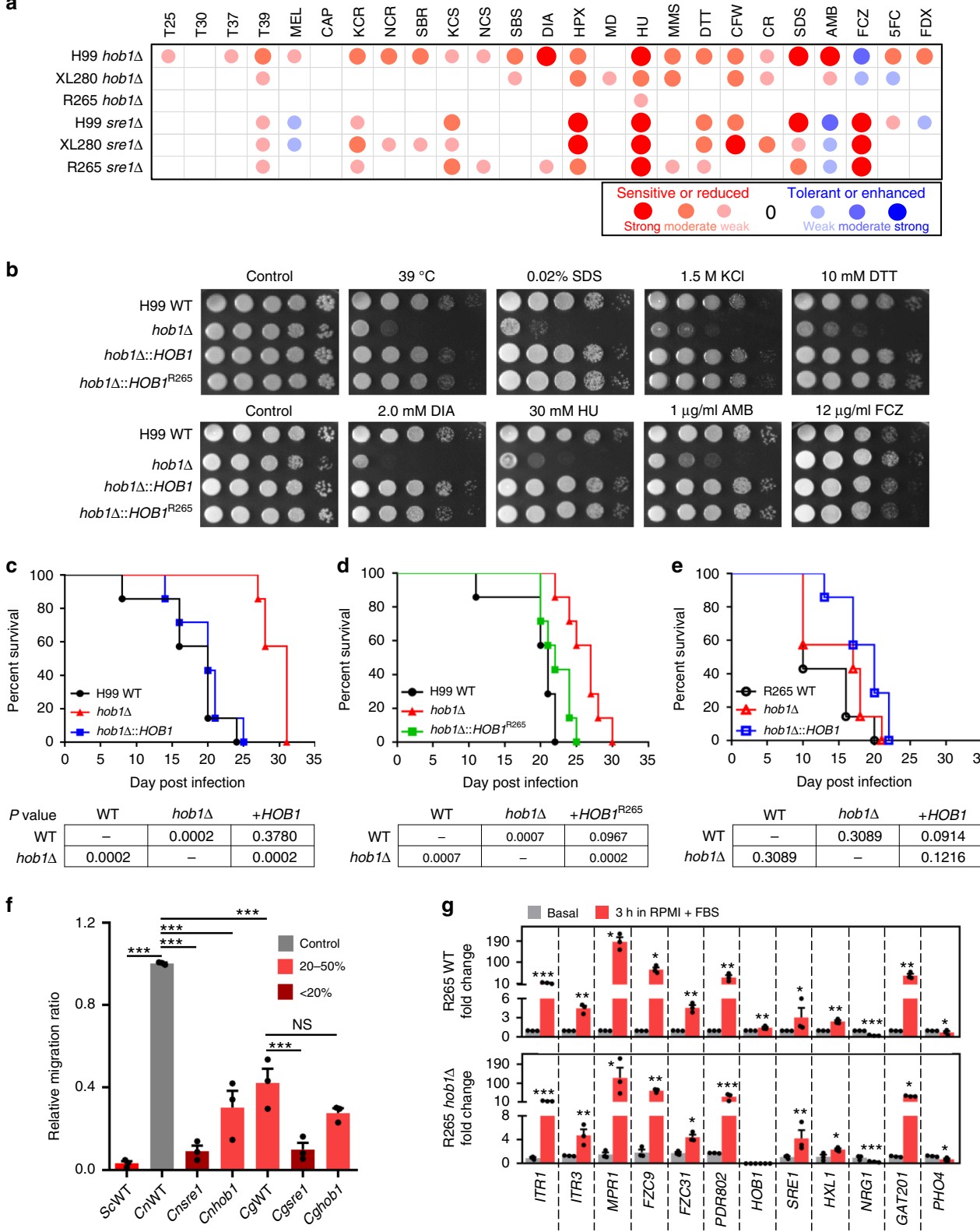

of the H99 *hob1*Δ mutants (Fig. 6b and Supplementary Fig. 9b). Surprisingly, the R265 *HOB1* allele restored all of WT phenotypic traits in the H99 *hob1*Δ mutants (Fig. 6b), indicating that Hob1 orthologues can work interchangeably in both strains. Collectively, these results indicate that Hob1-downstream targets, not Hob1 *per se*, are functionally distinct between H99 and R265 strains.

We next addressed the role of Hob1 in virulence of *C. neoformans* H99 and *C. deuterogattii* R265 strains using a murine (C57BL/6) intranasal instillation. As expected, the H99 *hob1*Δ mutant exhibited significantly attenuated virulence compared to WT and its complemented strain (*hob1*Δ::*HOB1*) (Supplementary Fig. 10a). In contrast, the R265 *hob1*Δ mutant showed rather enhanced virulence compared to the WT and its complemented

**Fig. 6 Hob1 plays pleiotropic roles in growth and pathogenicity of *C. neoformans*, but not in *C. deuterogattii*. a** Phenotypic heatmap of *hob1Δ* and *sre1Δ* mutants constructed in *C. neoformans* H99, *C. deneoformans* XL280, and *C. deuterogattii* R265 strain backgrounds based on data presented in Supplementary Fig. 8, which shows one representative spot assay data of more than three independent experiments. Each abbreviation are defined as follows: T25, T30, T37, and T39: growth rates at 25 °C, 30 °C, 37 °C, and 39 °C; MEL: melanin production levels; CAP: capsule production levels; KCR: YPD + KCl; NCR: YPD + NaCl; SBR: YPD + sorbitol; KCS: YP + KCl; NCS: YP + NaCl; SBS: YP + sorbitol; DIA: diamide; HPX: hydrogen peroxide; MD: menadione; HU: hydroxyurea; MMS: methyl methanesulphonate; TM: tunicamycin; DTT: dithiothreitol; CFW: calcofluor white; CR: Congo red; SDS: sodium dodecyl sulphate; AMB: amphotericin B; FCZ: fluconazole; 5FC: 5-flucytosine; FDX: fludioxonil. **b** H99 WT, *hob1Δ* mutant, and *HOB1* complementary and allele-swapping strains with H99 *HOB1* or R265 *HOB1* alleles were spotted onto YPD plates containing the indicated concentration of each chemical agent and photographed after 2–3 days. These spot assays were repeated more than three times, and one representative image was shown here. **c–e** Murine survival assay. A/J mice were infected by intranasal instillation with (**c**) *C. neoformans* WT (H99), *hob1Δ*, and *hob1Δ::HOB1* strains constructed in H99 strain background, (**d**) *C. neoformans* WT (H99), *hob1Δ*, and *hob1Δ::HOB1*[R265] strains constructed in H99 strain background, or (**e**) *C. deuterogattii* WT (R265), *hob1Δ*, and *hob1Δ::HOB1* strains constructed in R265 background. Survival was monitored for 35 dpi and statistical significance was determined using the Mantel–Cox test by comparing a pair of indicated groups (7 mice per group, n = 7). **f** BBB crossing assay of *hob1Δ* and *sre1Δ* mutants in *Cn* (H99) and *Cg* (R265). The *Sc* (*S. cerevisiae* S288C) strain was used as a negative control. Error bar indicates SEM for the three-independent experiments (n = 3), and statistics were calculated by one-way ANOVA with Bonferroni's correction. **g** HMC-mediated induction of brain-infection-related genes in *C. deuterogattii* WT (R265) and *hob1Δ* mutant as described in Fig. 5e. We performed three independent experiments (n = 3) with duplicates. Statistical significance of difference between basal and HMC condition was calculated by two-tailed Student's *t* test (unpaired). Asterisks indicate the following range of *P* values (\*, *P* < 0.05; \*\*, *P* < 0.001; \*\*\*, *P* < 0.0001).

strains (Supplementary Fig. 10b). However, deletion of *SRE1* abolished virulence in both H99 and R265 strains, whereas complementation with the *SRE1* allele restored WT virulence in the *sre1Δ* mutant (Supplementary Fig. 10). Next, we further confirmed the distinct roles of Hob1 in the virulence of H99 and R265 strains using the A/J mouse model. Similar to the results obtained using the C57BL/6 mouse model, the H99 *hob1Δ* mutant exhibited weakly but significantly attenuated virulence compared to WT (*P* = 0.0002) and its *hob1Δ::HOB1* complemented strain (*P* = 0.0002) (Fig. 6c). Notably, the *hob1Δ::HOB1*[R265] swapping strain exhibited wild-type levels of virulence (*P* = 0.0967) like the *hob1Δ::HOB1* complemented strain (Fig. 6d), further indicating that Hob1 orthologues can work interchangeably in both strains during in vivo host infection. In contrast, the R265 *hob1Δ* mutant showed WT virulence (Fig. 6e). These data indicate that Hob1 promotes the virulence of the *C. neoformans* H99 strain but not that of the *C. deuterogattii* R265 strain, while Sre1 plays a conserved role in the sibling *Cryptococcus* species complex.

To further examine the role of Hob1 in brain-infection of *C. neoformans* H99 and *C. deuterogattii* R265 strains, we monitored the ability of the *hob1Δ* mutants in both strain backgrounds to traverse the BBB. Reflecting the finding that *C. deuterogattii* does not target the brain during infection, the R265 strain did not cross the BBB as efficiently as the H99 strain, but traversed the BBB much more efficiently than *S. cerevisiae* (Fig. 6f). Deletion of *HOB1* markedly reduced the BBB crossing efficiency in the H99 strain, but not in the R265 strain (Fig. 6f). In contrast, *sre1Δ* mutants exhibited reduced the BBB crossing ability in both strains, further confirming the conserved role of Sre1. Surprisingly, expression of *ITR1* (*ITR1a* orthologue), *ITR3* (*ITR3c* orthologue), *MPR1*, *FZC9*, *FZC32*, *PDR802*, and *GAT201* was even more strongly induced by HMC in the R265 strain than in the H99 strain (Fig. 6g), indicating that these genes could not be responsible for discrepancy of the two *Cryptococcus* species. Notably, *PHO4* expression was not induced by HMC in the R265 strain, unlike the H99 strain in which it was reduced (Fig. 6g). Notably, deletion of *HOB1* did not affect the HMC-mediated induction of *ITR1*, *ITR3*, *MPR1*, *FZC9*, *FZC32*, *PDR802*, *SRE1*, and *GAT201*, and HMC-mediated reduction of *PHO4* (Fig. 6g) in the R265 strain. Collectively, the homeobox TF Hob1 play distinct roles in the pathogenicity of the *C. neoformans* H99 strain, but not in the pathogenicity of the *C. deuterogattii* R265 strain.

## Discussion

In this study, we systematically dissected complex signalling pathways that govern each stage of the infection process of *C. neoformans*, which causes fungal pneumonia and fatal meningoencephalitis at the early and late infection stages, respectively. Here, we identified a large number of signalling components that are involved in initial lung infection, haematogenous dissemination, BBB adhesion and crossing, and proliferation in the brain parenchyma of *C. neoformans*, allowing us to understand how this basidiomycete fungal pathogen utilises distinct and redundant sets of signalling pathways during the whole infection process. The list of kinases and TFs and their biological functions required for the infection process are summarised in Fig. 7 and Supplementary Data 2 and 3, based on the known information available in *C. neoformans* or other fungi.

The following processes are considered to be core-virulence functions that are required for the whole infection process of *C. neoformans*: nutrient sensing/metabolism, cell growth and development, fatty acid and phospholipid biosynthesis, cell wall/ membrane integrity, inositol polyphosphate signalling, stress responses, tRNA modification, redox reaction and vacuole/protein sorting. Not surprisingly, based on our previous phenome database of the 111 infection-related TF/kinase genes[15,16], 24% (22 kinases and 5 TFs) required for the growth at mammalian body temperature were essential for the whole spectrum of pathogenicity of *C. neoformans* (Fig. 7). Furthermore, 49% (37 kinases and 17 TFs) and 36% (30 kinases and 10 TFs) involved in capsule and melanin productions, respectively, were found to be core-virulence factors. Among the remaining genes, a majority of them were involved in stress response and adaptation. Therefore, our study further confirms that these virulence factors are critical for the whole infection processes of *C. neoformans*. As deletion of 12 kinases (Crk1, Dak202a, Fpk1, Irk1, Irk2, Irk3, Irk7, Mpk2, Psk201, Rim15, Snf101, and Trm7) and 16 TFs (Bwc2, Ddt1, Fzc17, Fzc22, Fzc38, Fzc5, Hel2, Jjj1, Mal13, Pdr802, Stb4, Yap4, Zfc1, Zfc2, Zfc3, and Znf2) did not result in any evident in vitro phenotypes, their essential in vivo functions need to be further elucidated in future.

Kinases and TFs, whose deletion resulted in reduced lung-STM scores but normal brain-STM scores, could play specific roles in lung infection (Fig. 7). Among these, the *fpk1Δ* mutant exhibits the lowest lung-STM score. However, deletion of *FPK1* does not result in any phenotypic changes, although its overexpression suppresses defective growth, stress resistance, and cell wall/ membrane integrity of the *ypk1Δ* mutant, suggesting that Fpk1

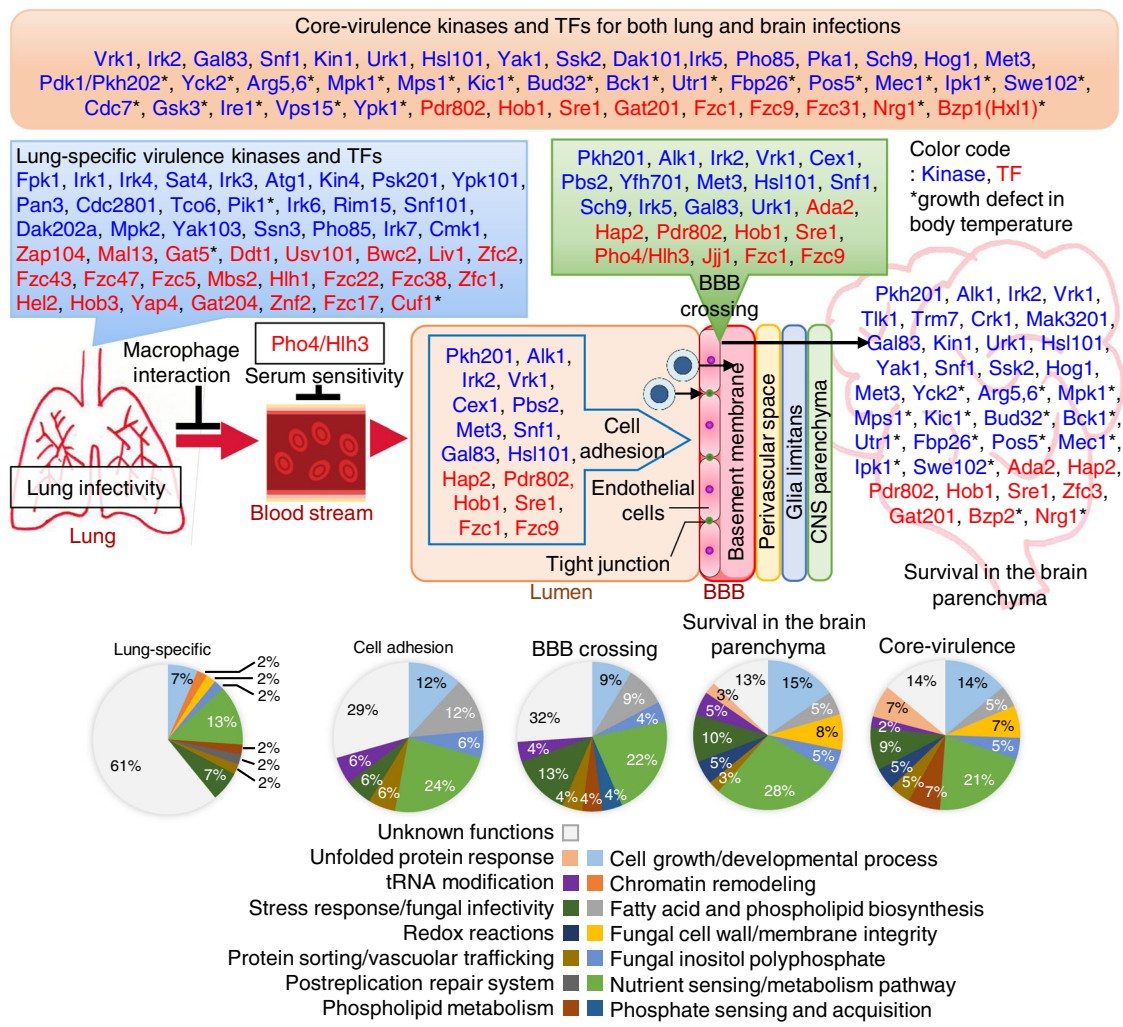

**Fig. 7 The signalling network regulating brain infection in *Cryptococcus neoformans*.** Gene Ontology (GO) term-based categorisation of lung-infection-related, brain-infection-related, and core-virulence kinases and TFs, which are summarised in Supplementary Data 2 and 3, are illustrated. Asterisk (*) indicates gene deletion mutants that exhibit growth defects at mammalian body temperature. Blue and red letters denote kinases and TFs, respectively, involved in the virulence of *C. neoformans*. Bold letters indicate kinases and TFs that play important roles in both BBB crossing and survival in brain parenchyma.

may have a role downstream of Ypk1[15]. Ypk1 is known to modulate the plasma membrane integrity in the TORC2 signalling pathway in *S. cerevisiae*[37]. In *C. neoformans*, we found that Ypk1 was required for both lung and brain infections. Therefore, Fpk1 could be responsible for a subset of Ypk1-dependent functions, which are likely to promote the lung infection of *C. neoformans*. Due to its potential role in maintaining the plasma membrane integrity, Fpk1 may be involved in the interaction of *C. neoformans* with lung alveolar macrophages. Our preliminary data demonstrate that Fpk1 indeed promotes inhibition of phagosome maturation within macrophages phagocytising *C. neoformans* (unpublished data).

Here, we found that *C. neoformans* utilises complex signalling pathways involved in diverse biological processes to infect the brain as summarised in Fig. 7. For BBB crossing, the following biological processes are involved: cell cycle regulation (Alk1 and Hsl101), tRNA export (Cex1), endoplasmic membrane assembly (Yfh701), stress response and adaptation (Pbs2), lipid and sterol metabolism and related regulations (Pkh201, Sre1, and Hob1), haeme-mediated respiration control (Hap2), ribosome biogenesis (Vrk1, Sch9 and Jjj1), carbon source utilisation (Gal83 and Snf1), methionine metabolism (Met3), histone acetylation and capsule

biosynthesis (Ada2), phosphate sensing and metabolism (Pho4), Tor signalling (Sch9), and unknown functions (Irk2, Irk5, Fzc1, Fzc9, and Pdr802). For survival in the brain parenchyma, the following biological processes are involved: glucose sensing and metabolism (Snf1, Gal83, Yck2, Fpb26, Yak1, and Nrg1), amino acid metabolism (Arg5,6 and Met3), polarised exocytosis (Kin1), pyrimidine ribonucleotide salvage pathway (Urk1), mitochondrial redox reactions (Pos5 and Utr1), tRNA modification (Trm7 and Bud32), replication or maintenance of double stranded RNA-containing particles (Mak3201), meiosis activation (Crk1, yeast Ime2 ortholog), cell cycle and morphology regulation (Mec1, Mps1, Alk1, Swe102, and Hsl101), stress response and adaptation (Ssk2 and Hog1), lipid-mediated regulations (Pkh201), histone acetyltransferase activity (Ada2), haeme-mediated respiration control (Hap2), ribosome biogenesis (Vrk1), Inositol polyphosphate biosynthetic process (Ipk1), sterol biosynthesis pathway (Sre1 and Hob1), cell wall/membrane integrity (Kic1, Bck1 and Mpk1), virulence factor regulation (Gat201) and unknown functions (Bzp2, Tlk1, Zfc3, Irk2, and Pdr802). Among these, 4 kinases (Pkh201, Alk1, Irk2 and Vrk1) and 5 TFs (Ada2, Hap2, Sre1, Hob1, and Pdr802) were required for both BBB crossing and survival in the brain parenchyma, indicating that lipid-mediated

regulation, cell cycle regulation, chromatin remodelling, and haeme-mediated respiration control are critical for both BBB crossing and survival inside the brain. Collectively, these findings indicate that *C. neoformans* employs complex signalling networks governing diverse biological processes to promote BBB crossing and survival/proliferation in the brain parenchyma.

Previously known brain infection factors include two inositol transporters (Itr1a and Itr3c)[38], hyaluronic acid synthase (Cps1)[39], metalloprotease (Mpr1)[31], urease (Ure1)[40,41], phospholipase (Plb1)[42], Fnx1[43], and Rub1[43]. Liu et al. demonstrated that addition of external inositol induces expression of *CPS1* in *C. neoformans* and enhances production of hyaluronic acid that facilitates the binding of the pathogen to CD44 glycoprotein in HBMECs[38]. In support, our in vivo transcriptional profiling analysis showed that expression of *ITR1A*, *ITR3C*, *MPR1*, *CPS1*, *PLB1*, and *FNX1* was indeed upregulated during brain infection, although their expression was often induced in other infected tissues (lungs, kidneys, and spleens). In addition to these known brain-infection factors, we found that redundant and distinct sets of kinases and TFs are involved in each step of brain infection of *C. neoformans*: adhesion to HMBECs, traversing the BBB, and proliferation in the brain parenchyma. Biological functions of genes involved in BBB adhesion and crossing are highly correlated and include nutrient sensing/metabolism, fatty acid/phospholipid biosynthesis, vacuolar trafficking, and inositol signalling. Notably, fatty acid/phospholipid biosynthesis and vacuolar trafficking are more important for BBB adhesion and crossing than for survival in the brain parenchyma. In contrast, genes involved in nutrient sensing and metabolism appeared to be most important for survival in the brain parenchyma, which supports the previous finding that the non-fermentative gluconeogenic pathway and the fermentative glycolytic pathway are required for the initial lung infection and the subsequent brain infection, respectively, of *C. neoformans*[44]. Besides those required for the growth at 37 °C, lung infection or BBB adhesion/crossing, several kinases and TFs (Tlk1, Trm7, Crk1, and Mak3201) appeared to be specifically required for the survival in the brain parenchyma (Fig. 7). Notably, Tlk1 encodes a Tor-like kinase, but its biological functions remain unclear. Although orthologous to Tor1, it does not bind to rapamycin and its deletion does not result in any evident in vitro phenotypic changes[15,45]. As Tor1 is involved in nutrient sensing pathway, it is conceivable that Tlk1 may play a role in sensing some brain-specific nutritional condition for *C. neoformans*.

Among the brain-infection-related kinases/TFs, four kinases (Pkh201, Alk1, Irk2, and Vrk1) and 4 TFs (Hap2, Pdr802, Sre1, and Hob1) were found to be required for all three brain infection steps (BBB adhesion/crossing and survival in the brain parenchyma), strongly indicating that they are key regulators of *C. neoformans* brain infection. Notably, here we found that Hob1 controlled the HMC-induced expression of *ITR1A*, *ITR3C*, and *MPR1* as well as other brain-infection-related TFs, including *FZC31*, *PDR802*, *SRE1*, and *PHO4*, further confirming that the homeobox TF is a critical transcriptional regulator of *C. neoformans* brain infection. The master regulatory role of Hob1 in brain infection of *C. neoformans* was further manifested by its dispensable role in *C. deuterogattii* (R265), which mainly targets the lungs instead of brain[33]. Our data showed that both H99 and R265 strains are able to cross the BBB, although the former strain does more efficiently than the latter strain. Notably, deletion of *HOB1* significantly impaired the ability of the H99 strain, but not the R265 strain, to cross the BBB. Our allele-swapping experiments demonstrated that the difference in Hob1 sequence was not responsible for the distinct roles of Hob1 in the two pathogenic *Cryptococcus* strains. There could be several explanations for this finding. Hob1 downstream target proteins may function

differently in the two species. Another possibility is that a Hob1-binding element could be differentially distributed in the promoter regions of Hob1 downstream targets. All these possibilities could be addressed in future RNA-seq and chromatin IP-seq analyses.

The current analyses have the following limitations. First, we could not track down the genes involved in neurotropism of *C. neoformans*, due to a lack of proper known systems. It has been hypothesised that *C. neoformans* has a neurotropism to the CNS to utilise neurotransmitters[13]. Therefore, some of the kinase and TF mutants may have reduced brain-STM scores because of a lack of neurotropism. Second, the in vitro HBMECs-based BBB transwell system used in this study is distinct from the natural, dynamic BBB structure, which consists of HBMECs, pericytes, and brain astrocytes[12]. Therefore, some of the kinase and TF mutants could be defective in crossing the natural BBB. The role of these signalling components in brain infection should be further dissected in future studies. Regardless of these limitations, this study allows us to understand complex signalling networks governing the whole infection processes of *C. neoformans* and provide therapeutic targets for treating fungal meningoencephalitis.

## Methods

**Ethics statement**. Animal care and all experiments were conducted in accordance with the ethical guidelines of the Institutional Animal Care and Use Committee (IACUC) of Yonsei University and National Taiwan University. The Yonsei University and National Taiwan University IACUC approved all of the vertebrate studies.

**The STM-based murine infectivity assay**. The high-throughput murine brain-infectivity test was performed using the previously published TF and kinase mutant libraries with the nourseothricin acetyltransferase (*NAT*) selection marker containing 46 unique signature tags (four and five groups of the TF and kinase mutant libraries, respectively)[15,16] (Supplementary Data 4). The *ste50Δ* mutants were used as virulent control strains, as previously reported[16,23]. In a given group of mutants with unique signature tags, each mutant was grown at 30 °C in YPD medium for 16 h separately and washed three times with phosphate-buffered saline (PBS). The concentration of each mutant was adjusted to $10^7$ cells per ml and 50 μl of each mutant sample was pooled into one tube. For preparation of the input genomic DNA, 200 μl of the mutant pool was spread on YPD plate, incubated at 30 °C for 3 days, and scraped. For preparation of the output genomic DNA samples, 50 μl of the mutant pool ($5 \times 10^5$ cells per mouse) was infected into seven-week-old female A/J mice (Japan SLC, Inc.) through intravenous or intracerebroventricular (ICV) injection. For intravenous injection, the tail was placed in warm (40 °C) water for expanding the vein and the tail vein injection was performed in restrained mice (3 mice per pooled mutant group). For ICV injection, mice were anaesthetised with 2% tribromoethanol by intraperitoneal injection (20 ml kg$^{-1}$, Sigma Aldrich) and placed on a stereotaxic device (David Kopf Instruments). The control strain and mutant pool were unilaterally injected into the ventricle (anteroposterior, −0.2 mm; lateral, −1.0 mm; ventral −2.0 mm) using a NanoFil needle (WPI) with Hamilton syringe and pump (WPI). The infected mice were sacrificed after 7 dpi and their brains were harvested. The recovered brains were homogenised in 5 ml PBS, 200 μl of them was spread onto the YPD plates containing 100 μg per ml of chloramphenicol, incubated at 30 °C for 2 days, and scraped. Total genomic DNA was extracted from scraped input and output samples by the cetrimonium bromide (CTAB) method. Quantitative PCR was performed with the tag-specific primers, as previously reported[15], using CFX96$^{TM}$ Real-Time PCR system (Bio-Rad). The STM score was calculated as previously described[16]. Briefly, relative changes in genomic DNA amounts were calculated by the $2^{-\Delta\Delta CT}$ method to determine the STM score. If the STM locus-tag was not detected in the output samples by qPCR, the amount of DNA was considered to be very small, and the maximum cycle of qPCR, 40, was used as the Ct value. The mean fold-changes in input versus output samples were calculated in Log score (Log$_2$ $2^{-(Ct,Target - Ct,Actin)output - (Ct,Target - Ct,Actin)input)}$). STM scores were measured for two independent mutant strains for each kinase and TF mutant (Supplementary Fig. 2).

**NanoString in vivo transcription profiling analysis**. Six-week-old female A/J mice were infected with $5 \times 10^5$ cells through nasal inhalation. Each group of 3 mice was sacrificed after 3, 7, 14 or 21 dpi. The lungs, brain, spleens and kidneys were recovered and lyophilised. Dried organs were homogenised and total RNA was extracted by using RNA extraction kit (easy-BLUE, Intron Biotechnology). Samples containing 10 ng of total RNA isolated from *C. neoformans* grown under in vitro basal conditions (30 °C; YPD medium) or 10 μg of total RNA isolated from

*C. neoformans*-infected mouse tissues were reacted with the designed NanoString probe code set and incubated at 65 °C overnight (12 to 18 h). The hybridised samples were processed on a NanoString prep station according to the manufacturer's standard protocol of the nCounter® platform, as previously reported[26,46]. Scanning was performed by digital analyser through high resolution (600 fields) option and normalised by nSolver software (provided by NanoString). A total of eight housekeeping genes were used for expression normalisation, including those used in a previous paper[47] (mitochondrial protein, CNAG_00279; microtubule binding protein, CNAG_00816; aldose reductase, CNAG_02722; cofilin, CNAG_02991; actin, CNAG_00483; tubulin β chain, CNAG_01840; tubulin α-1A chain, CNAG_03787; histone H3, CNAG_04828). The normalised data was transformed to $\log_2$ score to express the fold change and subject to clustering using one minus Pearson correlation with average linkage by Morpheus (https://software.broadinstitute.org/morpheus).

**Construction of gene deletion and complementation mutants.** The *mpr1Δ* mutant was reconstructed by double-joint PCR using *NAT*ᴿ-split marker and biolistic transformation (Supplementary Fig. 5), as previously described[15]. Briefly, 600 µg of gold microcarrier beads (0.6 µm, Bio-Rad) were coated with the PCR-amplified gene disruption cassettes and introduced into the *C. neoformans* cells by using particle delivery system (PDS-100, Bio-Rad). The transformed cells were further incubated at 30 °C for recovery of cell membrane integrity, scrapped after 4 h, and further incubated on YPD containing nourserothricin for selection. Hob1 orthologues in *C. deuterogattii* R265 (CNBG_5938) and *C. deneoformans* XL280 strains (CNC05440) and Sre1 orthologues in R265 (CNBG_9608) and XL280 (CNJ02310) were disrupted by double-joint PCR using *NAT*ᴿ-split marker and the primer sets presented in Supplementary Data 5 and biolistic transformation. Stable transformants were selected on YPD medium containing nourserothricin and screened by diagnostic PCR. The correct genotypes of screened mutants were confirmed by Southern blot analyses (Supplementary Fig. 6). To verify the phenotypes observed in *hob1Δ* and *sre1Δ* mutants of H99, R265 or XL280 strains, we constructed corresponding complementary strains (Supplementary Fig. 7) as previously described[48]. Briefly, PCR fragments containing the 5′- and 3′-flanking regions and the open reading frames of *HOB1* and *SRE1* genes in each background strain were amplified with species-specific primers (Supplementary Data 5). The amplified PCR products were cloned into the plasmid pTOP-V2 to confirm the DNA sequence. Due to the long length of each *SRE1* gene, we cloned the 5′-fragments of each *SRE1* gene using primer 1 and primer 2 (XmaI-SacII ends for H99, XbaI-Nhe1 ends for XL280, and XmaI-XhoI ends for R265) and the 3′-fragments using primer 3 and primer 4 (SacII-NotI ends for H99, Nhe1-ApaI ends for XL280, and XhoI-NotI ends for R265) listed in Supplementary Data 5. Each 3′-fragment was subcloned into the 5′-fragment in pTOP-V2, and their sequences were confirmed. The cloned whole genes were subcloned into the plasmid pJAF12 or pNEO vector containing neomycin-resistance gene (*NEO*) selection marker to produce plasmid pJAF12-HOB1 or pNEO-SRE1 in each species. The designated restriction enzyme-digested plasmid (AfeI for pJAF12-H99-HOB1, SwaI for pJAF12-R265-HOB1, MfeI for pJAF12-XL280-HOB1, AgeI for pNEO-H99-SRE1, NsiI for pJAF12-R265-SRE1, and HpaI for pNEO-XL280-SRE1) were linearised and biolistically delivered into the *hob1Δ* and *sre1Δ* mutants in each species. The re-integration of *HOB1* or *SRE1* into their native loci was confirmed by diagnostic PCR with specific primer sets presented in Supplementary Data 5. To construct the *HOB1* allele-swapping strain, the *C. deuterogatti* R265 *HOB1* ORF was inserted between *C. neoformans* H99 *HOB1* native promoter and terminator as follows (See schematic diagram in Supplementary Fig. 9b). The 5′-fragment (native promoter region of H99 *HOB1*) was amplified with specific primers, cloned into pTOP-V2 vector and sequenced. The 3′-fragment (R265 *HOB1* open reading frame with the native terminator region of H99 *HOB1*; NdeI-NsiI ends) was synthesized (Cosmo Genetech, Korea), cloned into pTOP-V2 vector, and then subcloned into the 5′-fragment (NotI-NdeI ends) in pTOP-V2. The whole insert was subcloned into the pNEO vector containing *NEO* selection marker to produce plasmid pNEO-HOB1ᴿ²⁶⁵. The plasmid was linearised by Afe1 and biolistically introduced into the *hob1Δ* mutant in H99. The *HOB1* substitution into their native loci (*hob1Δ::HOB1*ᴿ²⁶⁵) was confirmed by diagnostic PCR with specific primer sets (Supplementary Data 5).

**In vitro BBB crossing and adhesion assays.** Human brain microvascular endothelial cell (HBMEC) line (hCMEC/D3 cell line, Merck) was cultured performed as previously described[30,32]. Briefly, the $5 \times 10^4$ hCMEC/D3 cells in EGM™-2 medium (Lonza) were seeded on collagen (Corning) coated 8 µm-porous membranes (BD Falcon) for BBB crossing assay or 12-well plates (BD Falcon) for BBB adhesion assay. The day after seeding, medium was replaced with a fresh EGM™-2 medium supplemented with 2.5% human serum, and further grown for 4 days. A day before yeast inoculation, the medium was replaced with 0.5× diluted EGM™-2 medium and the cells were maintained at 37 °C and 5% $CO_2$ incubator. Integrity of tight junctions between hCMEC/D3 cells was confirmed by measuring the trans-endothelial electrical resistance (TEER), which should be around 200 Ω per cm². TEER was measured by Epithelial Volt per Ohm Metre (EVOM² device, World Precision Instruments). For BBB crossing assay, $5 \times 10^5$ cells of *C. neoformans* WT (H99), TF/kinase deletion mutants, or *S. cerevisiae* WT (S288C strain) were added to 500 µl of PBS and inoculated onto the top of the porous membranes. After 24 h

incubation at 37 °C in 5% $CO_2$ incubator, the number of yeast cells passing through the porous membrane was measured by counting CFU. TEER was measured by using EVOM² device before and after inoculation of yeast cells. The BBB migration ratio was calculated by dividing the output CFU of each tested strain by that of WT. For BBB adhesion assay, $5 \times 10^5$ yeast cells in 100 µl of PBS were inoculated onto a monolayer of hCMEC/D3 cells grown in a 12-well plate, and incubated for 24 h at 37 °C in 5% $CO_2$ incubator. Following incubation, cultures were washed 3 times with PBS and incubated with sterile distilled water for 30 min at 37 °C incubator to burst the hCMEC/D3 cells, and collected for CFU counting. The BBB adhesion ratio was calculated by dividing the adhered CFU of each test strain by that of WT *C. neoformans*.

**Expression analysis.** *C. neoformans* H99 strain and deletion mutants were incubated in YPD broth at 30 °C for 16 h, and each strain was sub-inoculated into 50 ml of fresh YPD broth and further incubated until the optical density of the medium at 600 nm ($OD_{600}$) reached the value of 0.8. The culture was separated into two-tubes (25 ml each), centrifuged, and washed three times with sterile distilled water. One tube was kept in liquid nitrogen tank to monitor basal expression levels, and the other tube was resuspended with equal volume of RPMI1640 medium containing 10% FBS. After 3 h incubation at 37 °C in a $CO_2$ incubator with 120 rpm horizontal shaking, cells were centrifuged and all samples were lyophilised overnight. Total RNA was extracted by using RNA extraction kit (easy-BLUE, Intron Biotechnology) and cDNA was synthesized using a Reverse Transcriptase kit (Thermo Scientific). Real-time quantitative PCR was performed for the target gene and *ACT1*; specific primer pairs are listed in Supplementary Data 5. The expression levels of the target gene were calculated by the threshold $2^{-\Delta\Delta CT}$ method.

**Growth and chemical susceptibility assays.** Cells grown overnight at 30 °C were serially diluted tenfold (1 to $10^4$ dilution) and spotted on YPD plate containing the indicated concentrations of chemical agents as follows: Osmotic (sorbitol) and cation/salt stress (NaCl and KCl) under either glucose-rich (YPD) or glucose-starved (YPD without dextrose; YP) condition; diamide (a thiol-specific oxidant), hydrogen peroxide ($H_2O_2$), menadione (as superoxide anion generator), and *tert*-butyl hydroproxide (an organic peroxide) for oxidative stress; methyl methanesulfonate (MMS) and hydroxyurea (HU) for genotoxic stress; tunicamycin (TM) and dithiothreitol (DTT) for ER stress; calcofluor white and Congo red for cell wall destabilising stress; sodium dodecyl sulfate (SDS) for membrane destabilising stress; amphothericin B, fluconazole, flucytosin, and fludioxonil for antifungal drug susceptibility. Cells were incubated at 30 °C and photographed during a period of 2–4 days. To test the growth rate of each mutant at distinct temperature, the serially tenfold diluted (1 to $10^4$ dilution) cells spotted onto the YPD plates were incubated at 25 °C, 30 °C, 37 °C, and 39 °C, and photographed as above.

**In vitro virulence-factor production assay.** To perform capsule production assay, cells were cultured in YPD broth at 30 °C overnight, and then spotted onto DME medium and incubated at 37 °C for 2 days. After incubation, the capsules were stained with India ink and visualised with an Olympus BX51 microscope equipped with a SPOT insight digital camera. For quantitative analysis of capsule production, 10% formalin-fixed cells were adjusted to $3 \times 10^8$ cells per ml in phosphate-buffered saline (PBS), and injected (50 µl) into microhaematocrit capillary tubes (Kimble Chase) in triplicates. All tubes were placed in an upright vertical position for 3 days to precipitate cells by gravity. The relative packed cell volume was measured with haematocrit capillary tubes by calculating the ratio of the lengths of the packed cell phase to the total phase (cells plus liquid phases), as previously described[15]. To examine melanin production, cells were grown overnight in YPD broth at 30 °C, then 5 µl of each culture was spotted on Niger seed media containing 0.1% or 0.2% glucose, incubated at 30 °C and photographed after 3–4 days.

**Murine virulence assay of *hob1Δ* and *sre1Δ* mutants.** Seven- to eight-week-old female C57BL/6 mice (BioLASCO Taiwan Co., Ltd) or A/J mice (Japan SLC, Inc.) were used to produce the murine intranasal infection model. *C. neoformans* or *C. deuterogattii* strains were cultured in YPD broth overnight at 30 °C with shaking at 200 rpm, and washed twice with PBS. Cells were counted with hemocytometer, resuspended in PBS, and diluted to $10^6$ cells (for C57BL/6 mice) or $10^7$ cells (for A/J mice) per ml. Appropriate dilutions of the cells were plated onto YPD agar plate and incubated at 30 °C for 48 h in order to confirm CFU numbers and viability. Groups of 10 (C57BL/6) or 7 (A/J) mice were anesthetised with isoflurane (Panion & BF Biotech Inc.) for C57BL/6 mice or 2% tribromoethanol (T48402, Sigma-Aldrich) for A/J mice, and inoculated with *C. neoformans* or *C. deuterogattii* strains via intranasal instillation of inocula of $5 \times 10^4$ cells (for C57BL/6 mice) or $5 \times 10^5$ cells (for A/J mice) in 50 µl PBS. The survival of mice was monitored twice daily for 49 days, and moribund mice (mice that were unable to eat or drink, their body weight was reduced by 15%, or were hunched) were euthanised with $CO_2$. The survival curve was created with Prism 8. The significance of differences was determined using Log-rank (Mantel-Cox) test.

**Statistical analysis.** Statistical analyses were performed with GraphPad Prism version 8. For the murine STM analysis, we used one-way analysis of variance (ANOVA) employing Bonferroni correlation to STM scores of *ste50Δ* (control) and

knockout mutants (cutoff > 2 or < −2) with three mice per STM group. In addition, the STM score of the second independent mutants was measured and calculated in the same manner in three independent mice. For the BBB crossing, adhesion, and gene expression assay, we used a two-tailed unpaired Student's $t$-test to compare the two groups. For the multiple comparison between the two-species of *C. neoformans* and *C. deuterogattii*, we used one-way ANOVA with Bonferroni's post correction. The murine survival assay was determined by using Log-rank (Mantel-Cox) test. For all analysis, $P < 0.05$ was considered significant and the error bars indicated ± standard error of the mean (SEM) from three or more independent experiments.

**Reporting summary**. Further information on research design is available in the Nature Research Reporting Summary linked to this article.

## Data availability

STM-based murine data for all kinase and TF mutants are provided in Supplementary Figs. 1 and 2. For the NanoString-nCounter® analysis, in vivo kinase and TF gene expression data and probe information are included in Supplementary Data 1. All the sequence and protein domain information was obtained from FungiDB (https://fungiDB.org) and UniProtKB (https://www.uniprot.org/), respectively. The source data underlying Supplementary Figs 5, 6, and 7 are provided as a Source Data file. Other data supporting the findings of this study are available from the corresponding authors upon request.

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

## Acknowledgements

This work was supported by National Research Foundation (NRF) grants funded by the Korean government (MSIT) (2016R1E1A1A01943365 and 2018R1A5A1025077 to Y.-S. B.; 2017R1A2B3011098 and 2017M3C7A1023471 to E.C; 2018R1C1B6009031 to K.-T.L.) and by the Strategic Initiative for Microbiomes in Agriculture and Food funded by Ministry of Agriculture, Food and Rural Affairs (918012-4 to Y.-S.B.). This work was also supported by the Korea Health Technology R&D Project through the Korea Health Industry Development Institute (KHIDI) funded by the Ministry of Health and Welfare (HI18C1664 to E.C.), the Brain Korea 21 (BK21) PLUS program, and by AmtixBio, Co., Ltd. (to Y.-S.B.).

## Author contributions

Y.-S.B. conceived the project. K.-T.L., J.H., D.-G.L., M.L., S.C., Y.-G.L., K.-W.J., A.H., Y.L., and S.-J.Y. performed experiments and analysed the data. K.-T.L., J.H., D.-G.L., Y.-L.C., J.-S.L., E.C., and Y.-S.B. supervised the experimental analysis. K.-T.L., J.H., D.-G.L., M.L., S.C., Y.-G.L., K.-W.J., A.H., Y.L., S.-J.Y., Y.-L.C., J.-S.L., E.C., and Y.-S.B. wrote the manuscript. All authors reviewed and approved this manuscript.

## Competing interests

Yonsei University and AmtixBio, Co., Ltd. have jointly filed a patent application (patent No. 10-2019-0114797), on which K.T.L., J.H., D.-G.L, J.-S.L., E.C., and Y.-S.B. are listed as inventors. Y.-S.B. and J.-S.L. are scientific co-founder and chief executive officer, respectively, of AmtixBio, Co., Ltd. E.C., Y.-S.B., and J.-S.L. are stock holders of AmtixBio, Co., Ltd. AmtixBio, one of the funders, played a role in the conceptualisation of this manuscript. All other authors declare no competing interests.
