## [Peer Review File · Nature Communications]

Reviewers' comments:

Reviewer #1 (Remarks to the Author):

Bahn's laboratory has created cryptococcal signature-tagged gene-deletion libraries (STM) of kinases and transcription factors (TFs) a couple of years ago and have used them to analyze signaling network systems pertinent to cryptococcal pulmonary as well as brain infection stages. In this study they present two independent reports: 1) comparison of STM based brain infectivity assay using IV infection route (systemic route) with those Lung STM data previously obtained by inhalation model; 2) analysis of in vivo transcription profiles of 180 TFs, 183 kinases, and 58 known virulence associated genes, in various organs during disseminated infection using NanoString technology.

It is a voluminous work and the first project identified core-virulence kinases and TFs required for both lung and brain infectivity as well as those of organ specific genes based on STM score. These are valuable new data. Deletion constructs of core-virulence kinases or TFs did not always result in changes in pathogenicity traits based on in vitro analysis which is not surprising. Second project concentrated on the expression profiles of numerous TFs, kinases and virulence associated genes during infection. The writing is clear and study purpose is well defined.

Minor comments

- Page 2. Line 35. The authors use the species name *C. deuterogattii* as if the isolate R-265 represent the species. R-265 is a strain of VGIIa type which has been extensively used for pathobiological study of *C. gattii* species complex. However, the VGII (*C. deuterogattii*) is one of the most genetically diverse species complex and R-265 is one of the many molecular types (i.e, VGIIa, b, n, x, etc) and known to be more diverse in transcriptional activity within *C. deuterogattii*. Use instead R-265 or a VGIIa strain of *C. deuterogattii* throughout the text.
- Page 4, line 76. Specify the strain H99 after *C. neoformans*.
- P. 7, line 132. ... during systemic murine infection can be confused with 'systemic infection model' which usually denote the infection via IV injection. To clarify, use 'during disseminated infection'
- P.7, line 145 .. YPD medium at 30oC. Is there any reason why 30oC was used instead of 37oC since the in vivo gene expression level will be clearly different from that of 30oC?
- P.8, Lines 152-157. The explanation on the expression levels of CAP genes need to be more specific. None of the CAP genes appear to be upregulated other than in the lung
- P.8, line 164. 14 dpi is not the initial stage of cryptococcal infection in the lung. Previous works have shown that after 7 days dpi, Cryptococci (H99) have already proliferated to reach more than one to 2 logs higher CFU than the day 1. By 2 weeks, the functions of recruited host immune cells are in full swing and should affect the cryptococcal gene expression. Thus, 14 dpi is already in a later stage of infection.
- P.8 line 171. Is there any reason why YPD (complex) media is used over defined media such as YNB or RPMI?
- P. 11 line 226. 14-21 dpi is defined as later stage (correct) and contradict with p8. Line 164.
- P.11, line 231. ... implying that... It is hard to compare in vivo data with in vitro data.
- P.13 to P. 14, line 293. Does this section belong to Results or discussion?
- P.14 line 302. Critical for brain infection... BBB crossing and brain infection are two different issues.
- P. line 303... Brain infection related.... Change into BBB-crossing related....
- P. 19. Line 418-419. How would Fpk1 affect crypto-macrophage interactions?
- P. 20, line 428...expression of ITR1,..... was indeed upregulated. Are they upregulated in HBMEC and in ICV?

Reviewer #2 (Remarks to the Author):

The authors performed a detailed analysis of the signaling networks that governs fungal brain infections using both in vitro and in vivo model systems. Overall, I can appreciate the amount of work invested into the wet lab experiments involving the murine and in vitro studies and the dry lab experiments which included a significant amount of data crunching. The data provided nice insight into the signaling networks governing brain infection by *C. neoformans* and also revealed a potentially important next line of inquiry (HOB1) that may reveal differences that control trafficking to the BBB that vary between *C. neoformans* and *C. gattii*. The studies and subsequent analysis are performed about as well as can be expected given the inherent limitations of these experiments (nicely identified by the authors). The data is well-described and placed into context.

Reviewer #3 (Remarks to the Author):

This is a well-written, comprehensive study, where dual signature-tagged transcription factor (TF) and kinase mutant libraries created by the Bahn laboratory (Lee KT et al Nat Commun. 2016 7:12766; Jung, K et al Nat Commun. 2015 6:6757), were used to systematically identify infection-stage dependent signalling pathways crucial for progression of disease from lung to the brain, the latter being fatal without treatment. The models used include a mouse inhalation and dissemination model, a newly established intracerebroventricular (ICV) brain infectivity model and a well-accepted human brain microvascular endothelial cell culture model of the blood brain barrier to measure fungal adherence and transmigration.

A key strength of the study is the use of several models to assess infectivity at each stage of infection. However, this is not well reflected in the abstract and should be. There is also no mention of use of nanostring technology (very novel in this context) to demonstrate that TF Hob1 is the master regulator for a number of brain-infection related genes in *C. neoformans*, but not in *Cryptococcus deuterogattii*. Finally, their key finding that Hob1 is the master regulator of brain-infectivity should be a key selling point in the abstract, which in its current form does not have a concluding statement. However, for a TF found to have such a prominent role in brain infectivity, its role in pathogenicity as assessed using the natural route of infection (i.e. inhalation model) would be expected to be more significant (see 1. below).

Comments and questions

1. Fig. 6. Given that HOB1 gene reconstitution in *C. neoformans* does not fully restore virulence to WT levels, has the HOB deletion mutant been created correctly? Were both the deletion mutant and the reconstitution strain validated by Southern blot to ensure absence of ectopic integrations? This is important because, once the median survival difference of the HOB1 reconstituted strain is taken into account, the difference in median survival for WT vs *hob1Δ* is reduced from approx. 15 to 8 days. 8 days difference in median survival time is not a huge difference given that Hob1 is concluded to be a master regulator of brain infectivity. Also why change the mouse strain from A/J (used in Lee KT et al Nat Commun. 2016 7:12766; Jung, K et al Nat Commun. 2015 6:6757) to C57BL 6, and reduce inoculum dose to 5×10^4 cells? Perhaps bigger differences in survival would have been seen with the A/J mouse strain?

2. Why the authors assess mutant growth in high serum media to identify mutants that are less likely to infect the brain due to inability to survive in blood, in lieu of obtaining blood culture STM scores directly in their mouse models, should be pointed out.

3. Line 56-57 BBB and proliferation in the brain parenchyma are critical factors for *C. neoformans* to impose lethal lesions on (within) mammalian brain tissues. Likewise, in a lot of other places "on" should be replaced with "in" or "within".

4. Line 347 What is a "shore N-terminal extension domain when referring to *Cryptococcus*

deuterogattii Hob1? Is "Shore" meant to mean real or definite?

5. Line 375 Check that "HMC" is defined after it is first used.

6. Line 242-243 Please explain why, in contrast to intracranial infection, the ICV injection allowed a more consistent and equal infection the brain parenchyma?

7. Fig. 7 legend. Should the last line read "survival in" brain parenchyma?

8. Line 394 The "followings" should read "following processes" or something similar.

[Responses to reviewers' comments]

Reviewer #1 (Remarks to the Author):

Bahn's laboratory has created cryptococcal signature-tagged gene-deletion libraries (STM) of kinases and transcription factors (TFs) a couple of years ago and have used them to analyze signaling network systems pertinent to cryptococcal pulmonary as well as brain infection stages. In this study they present two independent reports: 1) comparison of STM based brain infectivity assay using IV infection route (systemic route) with those Lung STM data previously obtained by inhalation model; 2) analysis of in vivo transcription profiles of 180 TFs, 183 kinases, and 58 known virulence associated genes, in various organs during disseminated infection using NanoString technology. It is a voluminous work and the first project identified core-virulence kinases and TFs required for both lung and brain infectivity as well as those of organ specific genes based on STM score. These are valuable new data. Deletion constructs of core-virulence kinases or TFs did not always result in changes in pathogenicity traits based on in vitro analysis which is not surprising. Second project concentrated on the expression profiles of numerous TFs, kinases and virulence associated genes during infection. The writing is clear and study purpose is well defined.

Response: We thank the reviewer for appreciating the quality of our work.

Minor comments

- Page 2. Line 35. The authors use the species name *C. deuterogattii* as if the isolate R-265 represent the species. R-265 is a strain of VGIIa type which has been extensively used for pathobiological study of *C. gattii* species complex. However, the VGII (*C. deuterogattii*) is one of the most genetically diverse species complex and R-265 is one of the many molecular types (i.e, VGIIa, b, n, x, etc) and known to be more diverse in transcriptional activity within *C. deuterogattii*. Use instead R-265 or a VGIIa strain of *C. deuterogattii* throughout the text.

Response: We agree with the reviewer. As suggested by the reviewer, we defined the strain R265 instead of just *C. deuterogattii* throughout the text.

- Page 4, line 76. Specify the strain H99 after *C. neoformans*.

Response: We agree. As suggested by the reviewer, we revised the text as follows.

Lines 73-76: Most importantly, we demonstrate that the homeobox TF Hob1 is the master regulator for a number of brain-infection related genes in the *C. neoformans* H99 strain, but not in the *Cryptococcus deuterogattii* R265 strain that does not target the brain during infection.

- P. 7, line 132. ... during systemic murine infection can be confused with 'systemic infection model' which usually denote the infection via IV injection. To clarify, use 'during disseminated infection'

Response: We agree. As suggested by the reviewer, we revised the subheading as follows.

Lines 131-132: In vivo gene expression profiling for *C. neoformans* TF and kinase genes during disseminated infection.

- P.7, line 145 .. YPD medium at 30oC. Is there any reason why 30oC was used instead of 37oC since the in vivo gene expression level will be clearly different from that of 30oC?

Response: We understand the reviewer's concern. The main reason is that we usually grow *Cryptococcus neoformans* strains at 30°C in YPD medium to prepare the initial inoculums for animal studies (fungal burden and survival assays). We acknowledge that multiple host-derived environmental factors should affect gene expression in *C. neoformans* during infection. These include temperature increase (from 25-30°C to 37°C), high CO₂ (5%), nutrient starvation, and host immune cells. Therefore, we decided to use 30°C as a basal condition. In future studies, we plan to further dissect individual host-derived signaling cues responsible for in vivo gene expression, which was out of the scope of the current study.

- P.8, Lines 152-157. The explanation on the expression levels of CAP genes need to be more specific. None of the CAP genes appear to be upregulated other than in the lung

Response: We agree with the reviewer. As suggested by the reviewer, we explained the in vivo expression patterns of each CAP gene more specifically as follows.

Lines 156-158: Unexpectedly, in vivo expression of CAP10, CAP59, CAP60, and CAP64 was generally downregulated in all infected tissues except the lungs where they were upregulated by about 2 fold between 7 to 21 dpi (Fig. 2b).

- P.8, line 164. 14 dpi is not the initial stage of cryptococcal infection in the lung. Previous works have shown that after 7 days dpi, Cryptococci (H99) have already proliferated to reach more than one to 2 logs higher CFU than the day 1. By 2 weeks, the functions of recruited host immune cells are in full swing and should affect the cryptococcal gene expression. Thus, 14 dpi is already in a later stage of infection.

Response: We agree with the reviewer. As suggested, we re-defined the infection stage as follows: 3 dpi (initial stage), 7 dpi (middle stage), and 14-21 dpi (late stages) throughout the manuscript (line 139-140; line 154-155). In this sentence, we intended to point out that transcriptional modulation of many pathogenicity-related kinases and TFs were initiated at the lung infection stage of cryptococcal infection. Thus, we revised the sentence as follows.

Lines 163-165: Collectively, these results indicated that transcriptional modulation of pathogenicity-related kinases and TFs appeared to be initiated at the lung infection stage of cryptococcal infection.

- P.8 line 171. Is there any reason why YPD (complex) media is used over defined media such as YNB or RPMI?

Response: We used YPD as a basal medium for measuring serum sensitivity because it is nutritionally rich and thus is not likely to impose any growth-related stresses (here the serum is the only variable). However, similar results were obtained when we compared RPMI+50% serum and RPMI medium.

- P. 11 line 226. 14-21 dpi is defined as later stage (correct) and contradict with p8. Line 164.

Response: We agree. As already suggested by the reviewer, we re-defined the infection stages as mentioned above.

- P.11, line 231. ... implying that... It is hard to compare in vivo data with in vitro data.

Response: We agree. We deleted the phrase and revised the sentence as follows.

Lines 230-232: In contrast, regardless of the critical roles of Cex1 and Met3 in BBB crossing and adhesion to HBMECs, their in vivo expression levels were not strongly induced at all infection stages and in all tissues (Fig. 3f).

- P.13 to P. 14, line 293. Does this section belong to Results or discussion?

Response: We agree. We moved the corresponding paragraph to the Discussion section.

- P.14 line 302. Critical for brain infection... BBB crossing and brain infection are two different issues.

Response: We agree. We revised the sentence as follows.

Lines 273-275: We found that expression of ITR1a, ITR3c, and MPR1 was strongly induced (~3-4 folds) by HMC (Fig. 5a), further indicating that inositol transporters and metalloprotease are critical for BBB crossing.

- P. line 303... Brain infection related.... Change into BBB-crossing related....

Response: We agree. We changed the sentence as suggested.

Line 275-277: We also monitored whether expression of any BBB-crossing related TFs that were discovered in this study were also induced by HMC.

- P. 19. Line 418-419. How would Fpk1 affect crypto-macrophage interactions?

Response: Regarding TFs and kinases whose deletion resulted in low STM score in the lungs, but not in the brain, we are currently investigating their role in interaction of *C. neoformans* with macrophage, in terms of phagocytosis, phagosome maturation, vomocytosis and pyroptosis efficiency. Because this is a big project in itself, we believe it is out of the scope of this paper, which is already lengthy, and plan to submit it as an independent manuscript. Having said that, our preliminary unpublished data clearly demonstrate that Fpk1 markedly promotes inhibition of phagosome maturation in *C. neoformans*. Therefore, we briefly described the role of Fpk1 in phagosome maturation as unpublished data as follows.

Line 404-406: Our preliminary data demonstrate that Fpk1 indeed promotes inhibition of phagosome maturation within macrophages phagocytizing *C. neoformans* (unpublished data).

- P. 20, line 428...expression of ITR1,..... was indeed upregulated. Are they upregulated in HBMEC and in ICV?

Response: The sentence refers to the NanoString-based *in vivo* transcriptional profiling data in brain tissues, which include both HBMEC and ICV. At this point, it is not clear whether *in vivo* expression of ITR1A, ITR3C, MPR1, CPS1, PLB1, and FNX1 is specifically upregulated only in HBMEC or ICV (or both). This issue should be further addressed in future studies.

Reviewer #2 (Remarks to the Author):

The authors performed a detailed analysis of the signaling networks that governs fungal brain infections using both *in vitro* and *in vivo* model systems. Overall, I can appreciate the amount of work invested into the wet lab experiments involving the murine and *in vitro* studies and the dry lab experiments which included a significant amount of data crunching. The data provided nice insight into the signaling networks governing brain infection by *C. neoformans* and also revealed a potentially important next line of inquiry (HOB1) that may reveal differences that control trafficking to the BBB that vary between *C. neoformans* and *C. gattii*. The studies and subsequent analysis are performed about as well as can be expected given the inherent limitations of these experiments (nicely identified by the authors). The data is well-described and placed into context.

Response: We thank the reviewer for appreciating the quality of our work.

Reviewer #3 (Remarks to the Author):

This is a well-written, comprehensive study, where dual signature-tagged transcription factor (TF) and kinase mutant libraries created by the Bahn laboratory (Lee KT et al Nat Commun. 2016 7:12766; Jung, K et al Nat Commun. 2015 6:6757), were used to systematically identify infection-stage dependent signalling pathways crucial for progression of disease from lung to the brain, the latter being fatal without treatment. The models used include a mouse inhalation and dissemination model, a newly established intracerebroventricular (ICV) brain infectivity model and a well-accepted human brain microvascular endothelial cell culture model of the blood brain barrier to measure fungal adherence and transmigration.

Response: We thank the reviewer for appreciating the quality of our work.

A key strength of the study is the use of several models to assess infectivity at each stage of infection.

However, this is not well reflected in the abstract and should be. There is also no mention of use of nanostring technology (very novel in this context) to demonstrate that TF Hob1 is the master regulator for a number of brain-infection related genes in *C. neoformans*, but not in *Cryptococcus deuterogattii*. Finally, their key finding that Hob1 is the master regulator of brain-infectivity should be a key selling point in the abstract, which in its current form does not have a concluding statement. However, for a TF found to have such a prominent role in brain infectivity, its role in pathogenicity as assessed using the natural route of infection (i.e. inhalation model) would be expected to be more significant (see 1. below).

Response: We agree that we did not fully describe the main findings of this study in the abstract section, mainly due to a word number limitation imposed by Nature Communications (approximately 150 words). As suggested by the reviewer, we revised the abstract as follows (total 165 words).

Abstract

Cryptococcus neoformans causes fatal fungal meningoencephalitis, but its complex signalling networks governing the infection process remain elusive. Here, we performed dual signature-tagged mutagenesis (STM)-based murine brain infectivity assay using transcription factor (TF)/kinase mutant libraries of the *C. neoformans* H99 strain in comparison with lung STM data and monitored in vivo transcription profiles of kinases and TFs during host infection using NanoString technology. These analyses identified many novel signalling components involved in blood-brain-barrier adhesion and crossing, or survival in the brain parenchyma. Among these, Pdr802, Hob1, and Sre1 were core virulence TFs required for all infection processes. Particularly, Hob1 controlled expression of several key brain-infection factors, including inositol transporters and a metalloprotease, as well as PDR802 and SRE1. Notably, however, Hob1 was dispensable for most cellular functions in the *Cryptococcus deuterogattii* R265 strain that does not target the brain during infection. In support, HOB1 deletion caused virulence defects in H99, but not in R265, indicating that Hob1 is a master regulator of brain-infectivity in *C. neoformans*.

Comments and questions

1. Fig. 6. Given that HOB1 gene reconstitution in *C. neoformans* does not fully restore virulence to WT levels, has the HOB deletion mutant been created correctly? Were both the deletion mutant and the reconstitution strain validated by Southern blot to ensure absence of ectopic integrations? This is important because, once the median survival difference of the HOB1 reconstituted strain is taken into account, the difference in median survival for WT vs *hob1* Δ is reduced from approx. 15 to 8 days. 8 days difference in median survival time is not a huge difference given that Hob1 is concluded to be a master regulator of brain infectivity. Also why change the mouse strain from A/J (used in Lee KT et al Nat Commun. 2016 7:12766; Jung, K et al Nat Commun. 2015 6:6757) to C57BL/6, and reduce inoculum dose to 5×10^4 cells? Perhaps bigger differences in survival would have been seen with the A/J mouse strain?

Response: The correct genotype for the *hob1* Δ mutant has been already confirmed by Southern blot analysis in our previous study (Jung, K et al Nat Commun. 2015 6:6757) and additionally confirmed by this study (Supplementary Figure 6). We verified the targeted integration of the wild-type HOB1 allele in the H99 and R265 *hob1* Δ mutants by using both diagnostic and Southern blot analysis (new Supplementary Figure 6). In support, all in vitro phenotypes of the *hob1* Δ ::HOB1 complemented strain were almost identical to those of wild-type strains (H99 and R265). However, it is still possible that the complemented strains may have some unidentified phenotypes and genotypes that are different from the wild-type strains, because multiple rounds of transformation during gene deletion and complementation may affect other parts of the genome. Therefore, here we re-constructed *hob1* Δ ::HOB1 complemented mutants and further confirmed the targeted re-integration of the wild-type HOB1 gene into its native locus through diagnostic PCR and Southern blot analysis (Supplementary Figure 6). In addition, the newly constructed *hob1* Δ ::HOB1 complemented mutants restored all wild-type phenotypes.

We agree with the reviewer that more dramatic results could have been obtained regarding the role of Hob1 in virulence if A/J mice and higher inoculum (5×10^5 cells) were used. Therefore, we performed the virulence assay once again using the standard A/J mouse model and *hob1* Δ mutant and newly constructed *hob1* Δ ::HOB1 complemented mutants. Furthermore, we independently tested the virulence recovery of the *hob1* Δ ::HOB1^{R265} swapping strain in the A/J mouse model. In these new

experiments, we decided not to include *sre1*Δ mutant sets, because the conserved role of Sre1 in virulence of *C. neoformans* H99 and *C. deuterogattii* R265 strains was very obvious in our previous data (complete avirulence). We moved the original Figure 6C and 6D to new Supplementary Figure 9 and added this new virulence assay data as Figure 6C-E. We found that the *hob1*Δ mutant in the H99 strain background, but not in the R265 strain background, still exhibited weakly but significantly reduced virulence even in the A/J mouse model, which is similar to the finding made in the C57BL/6 mouse model. Notably, the newly constructed *hob1*Δ::HOB1 complemented and *hob1*Δ::HOB1^{R265} swapping strains restored wild-type levels of virulence, further confirming our data. We revised the result and methods as follows.

Line 332-348: We next addressed the role of Hob1 in virulence of *C. neoformans* H99 and *C. deuterogattii* R265 strains using a murine (C57BL/6) systemic cryptococcosis model. As expected, the H99 *hob1*Δ mutant exhibited significantly attenuated virulence compared to WT and its complemented strain (*hob1*Δ::HOB1) (Supplementary Fig. 9a). In contrast, the R265 *hob1*Δ mutant showed rather enhanced virulence compared to the WT and its complemented strains (Supplementary Fig. 9b). However, deletion of SRE1 abolished virulence in both H99 and R265 strains, whereas complementation with the SRE1 allele restored WT virulence in the *sre1*Δ mutant (Supplementary Fig. 9). Next, we further confirmed the distinct roles of Hob1 in the virulence of H99 and R265 strains using the A/J mouse model. Similar to the results obtained using the C57BL/6 mouse model, the H99 *hob1*Δ mutant exhibited weakly but significantly attenuated virulence compared to WT (P=0.0002) and its *hob1*Δ::HOB1 complemented strain (P=0.0002) (Fig. 6c). Notably, the *hob1*Δ::HOB1^{R265} swapping strain exhibited wild-type levels of virulence (P=0.0967) like the *hob1*Δ::HOB1 complemented strain (Fig. 6d), further indicating that Hob1 orthologues can work interchangeably in both strains during in vivo host infection. In contrast, the R265 *hob1*Δ mutant showed WT virulence (Fig. 6e). These data indicate that Hob1 promotes the virulence of the *C. neoformans* H99 strain but not that of the *C. deuterogattii* R265 strain, while Sre1 plays a conserved role in the sibling *Cryptococcus* species complex.

Supplementary Figure 6. HOB1 and SRE1 genetic complementation. Knockout and complementation of HOB1 and SRE1 were confirmed by diagnostic PCR and Southern blot analysis. Specific primers for diagnostic PCR and Southern probes are listed in Supplementary Table 2. (a-f) HOB1 and SRE1 complementation in H99 *hob1*Δ and *sre1*Δ mutants. Schematic diagram of (a) HOB1 and (d) SRE1 in H99 WT, knockout mutant, and complemented strains. Targeted integration of HOB1 was confirmed by (b) diagnostic PCR and (c) Southern blot analysis with SpeI digestion. Ectopic integration of SRE1 was confirmed by (e) diagnostic PCR and (f) Southern blot analysis with BamHI digestion. (g-l) HOB1 and SRE1 complementation in R265 *hob1*Δ and *sre1*Δ mutants. Schematic diagram of (g) HOB1 and (j) SRE1 in R265 WT, knockout mutant, and complemented strains. Targeted integration of HOB1 was confirmed by (h) diagnostic PCR and (i) Southern blot analysis with StuI digestion. Targeted integration of SRE1 was confirmed by (k) diagnostic PCR and (l) Southern blot analysis with EcoRI digestion. (m-r) HOB1 and SRE1 complementation in XL280 *hob1*Δ and *sre1*Δ mutants. Schematic diagram of (m) HOB1 and (p) SRE1 in XL280 WT, knockout mutant, and complemented strains. Targeted integration of HOB1 was confirmed by (n) diagnostic PCR and (o) Southern blot analysis with SpeI digestion. Ectopic integration of SRE1 was confirmed by (q) diagnostic PCR and (r) Southern blot analysis with HindIII digestion.

2. Why the authors assess mutant growth in high serum media to identify mutants that are less likely to infect the brain due to inability to survive in blood, in lieu of obtaining blood culture STM scores directly in their mouse models, should be pointed out.

Response: We did not perform STM analysis of blood cultures directly recovered from mice infected with *C. neoformans* TF and kinase mutants, mainly because blood circulation is not a closed system. Previously, Shi et al. have reported that *C. neoformans* appeared in the brain microvasculature within 3-3.8 seconds after intravenous injection (Shi et al. J Clin Invest 2010). Therefore, once the signature-tagged *C. neoformans* mutants are hematogenously disseminated into other tissues, they are not equally distributed in blood cultures anymore. For example, their distribution in blood cultures will be affected by the number of mutants successfully colonizing brain as well as other tissues. Therefore, it is difficult to judge whether a mutant has low STM in mouse blood cultures simply because of the high

susceptibility to serum or other reasons. Therefore, to minimize such variables, we decided to use in vitro high serum media to address the question.

3. Line 56-57 BBB and proliferation in the brain parenchyma are critical factors for *C. neoformans* to impose lethal lesions on (within) mammalian brain tissues. Likewise, in a lot of other places “on” should be replaced with “in” or “within”.

Response: We agree. We checked all sentences including the words ‘brain parenchyma’ and revised the sentence as follows.

Lines 54-56: Among the infection stages, crossing the BBB and proliferation in the brain parenchyma are critical factors for *C. neoformans* to cause lethal lesions in mammalian brain tissues.

4. Line 347 What is a “shorter N-terminal extension domain when referring to *Cryptococcus deuterogattii* Hob1? Is “Shore” meant to mean real or definite?

Response: We apologize. It was a typo and should read “shorter”. We revised the sentence as follows.

Lines 322-323: R265 Hob1 with a shorter N-terminal extension domain (558 amino acids) exhibited 91.9% identity to H99 Hob1 (591 amino acids) (Supplementary Fig. 8a).

5. Line 375 Check that “HMC” is defined after it is first used.

Response: The abbreviation HMC was already defined at line 299 of the original manuscript: To this end, first we examined whether the known brain-infection factors are transcriptionally regulated by in vitro host-mimic conditions (HMC).

6. Line 242-243 Please explain why, in contrast to intracranial infection, the ICV injection allowed a more consistent and equal infection the brain parenchyma?

Response: We thank the reviewer for pointing this out. We meant to indicate that we injected *C. neoformans* into the brain reliably with a precise targeting method; we did not intend to compare our method to those of others. Therefore, we revised the sentence accordingly (line 240-242 in the revised version). In fact, most of the literature that we consulted regarding the injection of *C. neoformans* into the brain did not describe the details of intracranial injection sites or methods. Instead, most authors indicated that they injected *Cryptococcus* into the cerebrum. In this study, we injected the same number of *C. neoformans* cells into the ventricle with fixed coordinates (anteroposterior, -0.2 mm; lateral, -1.0 mm; ventral -2.0 mm) using a stereotactic tool as described in the Methods section, which let us infect the mice brain very consistently and reliably.

Line 240-242: For this purpose, we established an intracerebroventricular (ICV) method of infection of the mouse brain with *C. neoformans* by bypassing the BBB (Fig. 4a), which allowed us to infect the brain parenchyma consistently and with an equal number of cryptococcal cells.

Line 510-512: The control strain and mutant pool were unilaterally injected into the ventricle (anteroposterior, -0.2 mm; lateral, -1.0 mm; ventral -2.0 mm) using a NanoFil needle (WPI) with Hamilton syringe and pump (WPI).

7. Fig. 7 legend. Should the last line read “survival in” brain parenchyma?

Response: We agree. We have changed the sentence as suggested.

Lines 944-945: Bold letters indicate kinases and TFs that play important roles in both BBB crossing and survival in brain parenchyma.

8. Line 394 The “followings” should read “following processes” or something similar.

Response: We agree. We have changed the sentence as suggested.

REVIEWERS' COMMENTS:

Reviewer #3 (Remarks to the Author):

The authors have more than sufficiently addressed my queries. In particular, their modifications to the abstract now clearly highlight all of their significant findings and the methods used to obtain them. I also appreciate them recreating, and verifying construction of, a new reconstituted Hob1 mutant strain and repeating the virulence studies with this strain in mouse models using 2 mouse strains, to allow comparison to previous work published by the authors. I notice however that the Fig 6 legend stipulates use of "Intranasal instillation", when the text says "Systemic cryptococcus model". Please ensure methods in the 2 sections are consistent. The authors led by Prof Bahn, are to be highly commended on this significant and comprehensive body of work.

[RESPONSES TO REFEREES]

Reviewer #3 (Remarks to the Author):

The authors have more than sufficiently addressed my queries. In particular, their modifications to the abstract now clearly highlight all of their significant findings and the methods used to obtain them. I also appreciate them recreating, and verifying construction of, a new reconstituted Hob1 mutant strain and repeating the virulence studies with this strain in mouse models using 2 mouse strains, to allow comparison to previous work published by the authors. I notice however that the Fig 6 legend stipulates use of "Intranasal instillation", when the text says "Systemic cryptococcus model". Please ensure methods in the 2 sections are consistent. The authors led by Prof Bahn, are to be highly commended on this significant and comprehensive body of work.

Response: We appreciate that the reviewer acknowledged the efforts we made for this revised manuscript. Regarding the reviewer's final comment, we used three different murine infection methods in this study: 1) intranasal instillation, 2) intravenous (IV) injection, and 3) intracerebroventricular (ICV) injection. Intranasal instillation mimics the natural route of *C. neoformans* infection: the upper respiratory tract → the lower respiratory tract → hematogenous dissemination → brain and other organs. Intravenous injection of *C. neoformans* bypasses the respiratory tract to directly cause hematogenous dissemination and systemic infection. Both intranasal instillation and IV injection cause systemic cryptococcosis. In contrast, we directly infects the brain parenchyma and causes acute brain infection. The difference between IV and ICV injections methods was clearly stated in the Method section 2.

However, to avoid the confusion, we used the term "intranasal instillation" in the main text like the Figure 6c legend as suggested by this review. We revised the statement as follows.

Lines 429-430: We next addressed the role of Hob1 in virulence of *C. neoformans* H99 and *C. deuterogattii* R265 strains using a murine (C57BL/6) intranasal instillation.

Figure 6. (c-e) Murine survival assay. A/J mice were infected by intranasal instillation with (c) *C. neoformans* WT (H99), *hob1*Δ, and *hob1*Δ::HOB1 strains constructed in H99 strain background, (d) *C. neoformans* WT (H99), *hob1*Δ, and *hob1*Δ::HOB1^{R265} strains constructed in H99 strain background, or (e) *C. deuterogattii* WT (R265), *hob1*Δ, and *hob1*Δ::HOB1 strains constructed in R265 background.

Furthermore, wherever we used the term "systemic cryptococcosis", we clarify the infection methods. For example,

Figure 2. In vivo expression profiling of *C. neoformans* transcription factor and kinase genes in comparison with known cryptococcal virulence genes. (a) Graphical abstract of NanoString™-nCounter® based in vivo gene expression analysis in a murine model of systemic cryptococcosis through intranasal instillation.